# When are dynamical systems learned from time series data statistically accurate?

**Jeongjin (Jayjay) Park**
School of Computational Science and Engineering
Georgia Institute of Technology
Atlanta, GA 30332
`jpark3141@gatech.edu`

**Nicole Tianjiao Yang**
Department of Mathematics
Emory University
Atlanta, GA 30322
`tianjiao.yang@emory.edu`

**Nisha Chandramoorthy**[*]
Department of Statistics
The University of Chicago
Chicago, IL 60637
`nishac@uchicago.edu`

## Abstract

Conventional notions of generalization often fail to describe the ability of learned models to capture meaningful information from dynamical data. A neural network that learns complex dynamics with a small test error may still fail to reproduce its *physical* behavior, including associated statistical moments and Lyapunov exponents. To address this gap, we propose an ergodic theoretic approach to generalization of complex dynamical models learned from time series data. Our main contribution is to define and analyze generalization of a broad suite of neural representations of classes of ergodic systems, including chaotic systems, in a way that captures emulating underlying invariant, physical measures. Our results provide theoretical justification for why regression methods for generators of dynamical systems (Neural ODEs) fail to generalize, and why their statistical accuracy improves upon adding Jacobian information during training. We verify our results on a number of ergodic chaotic systems and neural network parameterizations, including MLPs, ResNets, Fourier Neural layers, and RNNs.

## 1 Introduction

Learning a dynamical system from time series data is a pervasive challenge across scientific domains. Such data come from expensive experiments and high-fidelity numerical models that simulate the underlying nonlinear, often chaotic, processes. The learning challenge is to train on available data to produce output models that i) provably preserve known symmetries and invariances (e.g., conservation principles); and ii) are inexpensive surrogates for use in downstream computations such as optimization and uncertainty quantification. The field of physics-guided machine learning [vdGSB+20, FO22, LK22, KKLL21, RPK19, KKL+21] has emerged in response, rapidly integrating neural networks into data-driven modeling and prediction workflows for a wide variety of complex dynamics, from geophysical fluid flows to phase transitions in materials (see [KMA+21, CCC+19] for surveys). Yet rigorous generalization analyses of neural parameterizations in these applications, wherein the underlying dynamics can exhibit chaotic behavior, have been underexplored.

---

[*]Corresponding author

38th Conference on Neural Information Processing Systems (NeurIPS 2024).

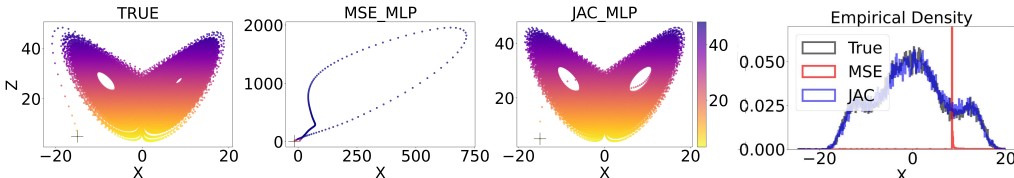

Figure 1: A random orbit on the x-z plane obtained from RK4 integration of the Lorenz vector field ([Lor63] (first column), Neural ODE, 'MSE_MLP', trained with mean-squared loss (second column) and Neural ODE, 'JAC_MLP', trained with Jacobian loss (third column). The last column shows the empirical PDF of the orbit generated by the true (gray), 'MSE_MLP' (red) and 'JAC_MLP' (blue) models. Experimental settings and additional results are in Appendices B and C respectively. **Gist:** A model trained well with MSE can produce atypical orbits but when Jacobian information is added to the training, it reproduces the long-term/statistical behavior accurately.

Here we investigate data-driven neural parameterizations of chaotic ODEs/PDEs and maps (discrete-time dynamical systems) motivated by the following observation: a neural representation that is learned well, i.e., with a small generalization error, can still produce *unphysical* long-term or ensemble behavior. Figure 1 (left) shows the classical Lorenz '63 chaotic attractor [Lor63] plotted using a long random trajectory/*orbit* (time integration of the Lorenz equation vector field starting from a random initial condition, which is indicated by a '+' sign). The second column shows an orbit from a neural network model, 'MSE_MLP', which minimizes the mean-squared error in the represented vector field at 10,000 training points and shows high accuracy ($< 5\%$ average relative error) over 8000 test points on the attractor. Surprisingly, Figure 1(column 2) shows that the learned 'MSE_MLP' Neural ODE [CRBD18] model produces an atypical orbit – an orbit different from almost every orbit of the true system – for the same randomly chosen initial condition. As a result, the learned empirical distribution is not close to the physical distribution – that of almost every true orbit, as shown in Figure 1 (column 4). On the other hand, the 'JAC_MLP' model, which is obtained by minimizing the mean-squared error in the vector field and its first derivative (Jacobian matrix), reproduces the Lorenz '63 attractor and the physical distribution on the attractor (Figure 1, column 4). The 'JAC_MLP' also captures all the Lyapunov exponents (LEs) – measures of asymptotic stability to perturbations, which are invariants in ergodic systems – accurately, while the 'MSE_MLP' only obtains the leading LE accurately.

Naturally, we ask about the wider applicability of these observations. For any ground truth dynamical system, how can we quantify the probability of success, including obtaining sample complexity results, of learning its *physical* or typical behavior? That is, how do we redefine and extend generalization analyses to neural representations of complex dynamical systems? To answer these questions, we start with a deceivingly simple supervised learning setup: given $m$ samples from a time series, $\{(x_t, x_{t+1})\}_t$, $0 \leq t \leq (m-1)$, how can we learn a model, $F_{nn}$, such that, i) $x_{t+1} \approx F_{nn}(x_t)$, for all time $t$, and ii) the underlying distribution of the states $x_t$ and other dynamical invariants such as Lyapunov exponents are reproduced by orbits of $F_{nn}$? It is widely accepted that learning such an $F_{nn}$ involves matching orbits of $F_{nn}$ with $x_t$ over large $t$ during training. However, small errors propagate over orbits, by definition, in a chaotic system, leading to training instabilities. In response, a vast literature has been dedicated to developing sophisticated training models based on RNNs [PLH+17, PWF+18, RM21], operator learning [LLSK+22] and regularizations [LG22, FJNO20].

We focus instead on the empirical risk minimization (ERM) for $F_{nn}$ that does not explicitly use the temporal correlations/dynamical structure in the data, avoiding training instabilities. Thus, we attempt to characterize when an elementary regression problem $F_{nn}$ can still lead to learning a physical representation, leading to a practical theory of learning chaotic systems from data. Our specific contributions are as follows:

- Motivated by extensive empirical results, we develop useful notions of generalization that characterize a model's ability to reproduce dynamical invariants.

- We develop new dynamics-aware generalization bounds for minimization of errors in the $C^r$, $r = 0, 1$, topology.

- We leverage shadowing theory from dynamical systems to rigorously characterize failure modes in learning statistically accurate models.

## 2 Generalization of parameterized ergodic dynamics

In this section, we motivate, through illustrative examples, the need for a dynamics-aware definition of generalization in the context of learning from time series data. We introduce concepts from dynamical systems as needed for a self-contained presentation.

**Dynamics.**: A *map*, or a discrete-time dynamical system, $F$, is a function on a closed and bounded (compact) set, $M \subset \mathbb{R}^d$, which is called the state/phase space of the dynamics. We exclude scenarios where the dynamics can be unbounded, and focus on settings where $F \in C^1(M)$ is a differentiable function on $M$. We denote by $F^t$, $t \in \mathbb{Z}^+$, the iterates of the dynamics, or the compositions of $F$ with itself $t$ times, i.e., $F^t = F \circ F^{t-1}$. An *orbit* or trajectory of $F^t$ starting at an initial state $x \in M$ is the sequence $\{F^t(x)\}_{t \in \mathbb{Z}^+}$. In practice, $F$ may be a numerical ODE solver that approximates the continuous-time solutions, $\varphi^t, t \in \mathbb{R}^+$, of the ODE (written in dynamical systems notation[2] ): $d\varphi^t(x)/dt = v(\varphi^t(x))$, where $v : M \to TM$ is the true vector field describing the governing equations. Fixing some $\tau \in \mathbb{R}^+$, $F := \varphi^\tau$. We say a map $F$ is chaotic if there exists a subbundle of $TM$, called the unstable subbundle, where infinitesimal perturbations grow exponentially under the linearization (Jacobian map), $dF^t$, of the dynamics. The true map $F$ generates a deterministic, autonomous system (see Appendix A).

**Physical measures.** A probability measure $\mu : M \to \mathbb{R}^+$ is a *physical* measure [You02] for the dynamics $F$ if it is a) $F$-invariant, b) ergodic and c) *observable* through $F$. A measure $\mu$ is observable if time-averages along any orbit starting from a randomly chosen initial point converge to constants that are expectations (phase space average) with respect to $\mu$. That is, for any $f \in \mathcal{C}(M)$, $(1/T) \sum_{t \leq T} f(F^t(x)) \xrightarrow{T \to \infty} \mathbb{E}f(x)$, for any initial point $x$ chosen Lebesgue a.e. on a set $U \subseteq M$. The orbits starting almost everywhere on the basin of attraction, $U$, asymptotically enter a set, $\Lambda$, called the attractor. In dissipative chaotic systems, the attractor, $\Lambda$, which is the compact support of the physical measure $\mu$, has Lebesgue measure 0. Consequently, $\mu$ may not have a probability density, that is, $\mu$ is not absolutely continuous, or is singular, with respect to Lebesgue measure.

**Neural ODE.** Introduced in [CRBD18], a Neural ODE, denoted here by, $v_\theta : M \to \mathbb{R}^d$, with parameters, $\theta$, is a vector field represented by a neural network. The vector field can be time integrated to obtain solutions $\varphi_\theta^t : \mathbb{R}^d \to \mathbb{R}^d, t \in \mathbb{R}^+$ to the ODE, $d\varphi_\theta^t(x)/dt = v_\theta(\varphi_\theta^t(x))$. As noted in the introduction, suppose we have $n$ distinct pairs $S = \{(x_i, F(x_i))\}_{i \in [n]}$, which could come from a single orbit of $F$, as our training data. We train the Neural ODE by solving an ERM for the loss, $\ell$, that can be chosen to be a square loss, e.g., $\ell(x, \varphi_\theta^\tau) = \|\varphi_\theta^\tau(x) - F(x)\|^2$. That is, we solve for $\theta$ that minimizes the training loss, $(1/n) \sum_{x \in S} \ell(x, \theta)$. Note that the true ODE or vector field is not explicitly used in training, only the solution map at some time intervals, $F$. We refer to the map, $x \to \varphi^\tau(x)$, as $F_{nn}$, or neural representation of the target map, $F$.

**Data and optimization.** Suppose we use an $m$-length orbit as our training data, i.e., $x_{i+1} = F(x_i)$, then, the data are *not*, strictly speaking, independent. However, a feature of chaotic systems is an exponential decay of correlations, and so training data of the form, $\{(x_i, F^\omega(x_i))\}$ starting from an initial condition $x_0 \sim \mu$, can be thought of as iid, for a large enough $\omega \in \mathbb{N}$. In that case, the learned map $F_{nn}$ represents the function $F^\omega$. During optimization, infinitesimal linear perturbations, will have to be evolved for time $\omega$. Since adjoint solutions blow up exponentially, a longer $\omega$ will lead to training instabilities. To avoid numerical difficulties in training and focus on issues surrounding generalization, we choose $\tau := \delta t$, a small time step, to define the target map $F$ and learn a neural network representation of this function. When viewed this way, the generalization of Neural ODEs can be analyzed through the conventional lens of supervised learning with the loss,

$$\ell(x, F_{nn}) = \|F_{nn}(x) - F(x)\|^2, \tag{1}$$

where $F_{nn} := \varphi^{\delta t}$ is a neural network representing the map. For a map $h : M \to M$, we define the training and generalization errors in the usual way:

$$\hat{R}_S(h) = (1/m) \sum_{i=1}^m \ell(x_i, h), \quad R(h) = \mathbb{E}\ell(x, h), \tag{2}$$

---

[2]We use this notation rather than the customary $dx(t)/dt = v(x(t)), x(0) = x_0$, since in dynamical systems, we are interested in all possible orbits, as opposed to a particular orbit/path, for which defining the flow, $\varphi^t$, is necessary.

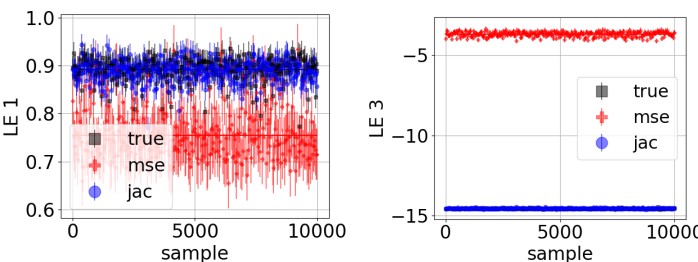

Figure 2: Learned and true LEs computed over 30,000 time steps using the QR algorithm of Ginelli et al [GCLP13] starting from 10,000 random initial states. The "MSE" and "JAC" labels indicate computations using the Neural ODE models trained with the loss functions in (1) and (3) respectively.

where the expectation is over the data distribution, which may be $\mu$ or any initial probability distribution of the states.

**Example.** To illustrate our conceptual findings surrounding generalization, we use the canonical Lorenz '63 system, which is a 3-variable reduced order model of atmospheric convection [Lor63]. The ODE can be written as $d\varphi^{t}(x)/dt = v(\varphi^{t}(x))$, where the vector field $v$ is given by $v(x) = [\sigma(\mathrm{y} - \mathrm{x}), \mathrm{x}(\rho - \mathrm{z}) - \mathrm{y}, \mathrm{xy} - \beta \mathrm{z}]^{\top}$, and $x = [\mathrm{x}, \mathrm{y}, \mathrm{z}]^{\top}$ are the coordinate functions of the state $x$. We use the standard values of the parameters $\sigma = 10, \rho = 28, \beta = 8/3$, at which the solutions are chaotic. We use the Runge-Kutta 4-stage time integrator with a time step size of $0.01$ to define the map $F$. That is, $F(x)$ is the solution $\varphi^{0.01}(x)$ approximated by an RK4 time-integrator. Our Neural ODE map, $F_{\mathrm{nn}}$, is learned to approximate $F$ by solving the above optimization with $n = 10,000$ training points along an orbit. We illustrate our numerical results on various Neural ODE models and architectures that have fully connected layers, ResNet blocks with convolutional layers, and Fourier neural operators [LKA+20]. Many such models learn accurate representations of the true vector field $v$, as evidenced by small training and test errors (sample average approximation of the generalization error in (2) over 8,000 points from the data distribution). These are shown in Figure 3 (Appendix C), while other hyperparameter and optimization settings are in Appendix B.

**Statistical measures and Lyapunov exponents.** Our numerical results test the accuracy of models beyond generalization error as defined in (2). In particular, we compute time-averages using the Neural ODE models and compare against expectations with respect to $\mu$ obtained from the true equations. For the Lorenz system, we note that time averages obtained from the models with small generalization errors can be inaccurate. That is, even if the errors in the vector field are small (see also Figure 4 in Appendix C), the time-averages can match poorly. This is described for the best performing Neural ODE model in Table 3. The discrepancy in the learned distribution is shown in terms of Wasserstein distance computed using empirical distributions on long orbits (of length 50,000). We find that the Neural ODE model does not learn the ground truth statistics even if the training data include transient dynamics off the attractor.

Lyapunov exponents, roughly speaking, measure the asymptotic exponential growth/decay of infinitesimal perturbations under the Jacobian map $dF$. In an ergodic system, they are independent of the initial state and can be written as an expectation with respect to $\mu$. In this work, Lyapunov exponents are yet another dynamical invariant (statistical quantity) in ergodic systems that we use to evaluate the statistical fidelity of learned models. In a chaotic system, there is at least one positive Lyapunov exponent, and the number of positive Lyapunov exponents is the dimension of the unstable manifold. The Lorenz system has one positive Lyapunov exponent (LE), one zero LE (corresponding to a center direction, or the vector field $v$ itself) and one negative LE. The ground truth map $F$, which is a time $\delta t$ approximation of the flow $\varphi^{t}$ has a two-dimensional center-unstable manifold and a one-dimensional stable manifold.

In Figure 2, we plot the Lyapunov exponents obtained using a classical QR iteration-based algorithm [GCLP13]. The Neural ODE model, marked 'MSE', produces a reasonably close approximation of the ground truth value ($\approx 0.9$) for the positive LE but obtains an incorrect approximation of the stable LE (true value $\approx -14.5$). We note also that standard deviations in the LE values computed over different orbits is quite large, indicating some orbits with atypical behaviors. We show the $\ell_2$ error in the computed LEs by the Neural ODE model in Table 3, and full details in Table 5.

Table 1: True vs. learned Lorenz system: comparison of statistics. ($W^1$: Wasserstein-1 Distance, $\Lambda$: set of LEs, $\hat{\mu}_T$: empirical distribution of an orbit $T$. The subscript NN indicates quantities computed using NN models trained with different loss functions (MSE (1), JAC (3)).

| | | Norm Difference | | |
|---|---|---|---|---|
| Model | Loss | $W^1(\hat{\mu}_{500}, \mu_{\text{NN},500})$ | $\|\Lambda - \Lambda_{\text{NN}}\|$ | $\|\langle x \rangle_{500} - \langle x \rangle_{500,\text{NN}}\|$ |
| MLP | MSE | 18.9711 | 9.6950 | 15.2220 |
| MLP | JAC | 0.6800 | 0.0118 | 0.6524 |
| ResNet | MSE | 1.3567 | 10.8516 | 0.7760 |
| ResNet | JAC | **0.1433** | **0.0106** | **0.0559** |
| FNO | MSE | 10.5409 | 22.1600 | 9.4270 |
| FNO | JAC | 1.3076 | 0.0505 | 0.9748 |

**Jacobian-matching.** We now consider Neural ODE models trained with the following loss function.

$$\ell_\lambda(x, F_{\text{nn}}) = \|F_{\text{nn}}(x) - F(x)\|^2 + \lambda \|dF_{\text{nn}}(x) - dF(x)\|^2, \tag{3}$$

where $F_{\text{nn}} : M \to M$ is a Neural ODE map, $dF_{\text{nn}}(x) : T_x M \to T_x M$ its ($d \times d$) Jacobian matrix at $x$, and $\lambda > 0$ is a hyperparameter that scales the relative importance of the two terms. We train in the usual way, by solving an ERM problem to minimize the training error, $\hat{R}_{S,\lambda}$, which is defined as before, but with the new Jacobian-matching loss:

$$\hat{R}_{S,\lambda}(h) = (1/m) \sum_{i=1}^{m} \ell_\lambda(x_i, h), \quad R_\lambda(h) = \mathbb{E}\ell_\lambda(x, h). \tag{4}$$

We train Neural ODE models with similar architectures as above, and find best performing models in terms of the test error (approximation of the generalization error) in (4). As in the training with the loss function $\ell$, we find several accurate neural representations of the Lorenz map (see training and test loss plots in Appendix C). Their performance on statistical measures and Lyapunov exponent predictions is remarkably different, however. In Figure 1, a Neural ODE trained with the Jacobian-matching loss produces an attractor (column 3) that is visually similar to the Lorenz attractor. The empirical distributions match closely with the ground truth, as shown in column 5 of Figure 1 and in Table 3, where the model is marked with a 'JAC', while one trained with the loss $\ell$ in (1) is indicated with an 'MSE'. The LEs and statistical averages of the state are all accurately represented, although the model is not explicitly designed to learn temporal patterns in the data. The MSE models learn vector fields with comparable accuracy with the Jacobian-matching models. That is, they learn accurate representations of $F$ with high probability (over the data distribution), but failing to learn $dF$ accurately leads to statistical inaccuracy. Due to the presence of atypical orbits, the attractor and the physical measure are not reproduced. Even though our formulation of Neural ODEs is simply stated as an ERM for regression, the generalization errors $R(F_{\text{nn}})$ and $R_\lambda(F_{\text{nn}})$ cannot determine whether $F_{\text{nn}}$ is a physical representation of the dynamics.

## 3 When generalization implies statistical accuracy

In the previous section, we observed that adding information about the Jacobian in the training process led to statistically-accurate learning. Does this observation apply more broadly to other ergodic systems? How can one further improve generalization? In this section, we provide answers to these questions by proving dynamics-aware generalization bounds for Neural ODEs.

Our ultimate goal is to minimize a statistical loss function rather than (1) or (3) since this measures how accurately a model $h$ can reproduce ergodic averages associated with $F$. For instance, $\ell^{\text{stat}}(x, h) = \sup_{f \in \text{Lip}_1} |\mathbb{E}[f(x)] - \lim_{T \to \infty} (1/T) \sum_{t \leq T} f(h^t(x))|$. This is the Wasserstein ($W^1$) distance (expressed in its dual form) between the ergodic measure $\mu$ and the ergodic measure associated with the orbit of $h$, a learned model, starting at $x$, $\lim_{T \to \infty} \text{Unif}\{x, h(x), h^2(x), \cdots, h^T(x)\}$. As noted, ERMs for this loss function do not have a straightforward implementation when the map $h$ sought is chaotic. Hence, we seek conditions under which solving ERMs defined with losses (1) and (3) can still minimize $\ell^{\text{stat}}$. To understand when solving regression problems for $F_{\text{nn}}$ lead to statistically accurate physical representations, we first assume that a notion of "shadowing" applies to $F$.

**Definition 1.** *We say that the shadowing property applies to a map $F$ if for any $\delta > 0$, there exists an $\epsilon = \mathcal{O}(\delta)$ so that for any map $G$ with $\|G - F\|_1 := \sup_{x \in M}(\|G(x) - F(x)\| + \|dG(x) - dF(x)\|) \leq \epsilon$, there exists a map $\tau : M \to M$ close to the identity such that $\|G^t(\tau(x)) - F^t(x)\| \leq \delta$ for all $t$.*

Intuitively, the shadowing property means that an orbit of a nearby dynamical system can closely follow a true orbit – called a shadowing orbit – of $F$ for all time. This kind of uniform-in-time shadowing is a classical result for a mathematically ideal class of chaotic systems, called uniformly hyperbolic systems (see Katok and Hasselblatt [KKH95] Ch 18; Appendix A). For a textbook presentation and extension of shadowing for dynamical systems with some hyperbolicity, see [Pil06]. For a uniformly hyperbolic $F$ (see Appendix A), we now assume that a neural representation $F_{\mathrm{nn}}$ of $F$ trained with $n$ samples generalizes well in terms of $C^1$-distance. That is, an ERM solution for the loss 3 ('JAC' models in sections 1 and 2) generalizes so that $R_\lambda(F_{\mathrm{nn}})$ is small.

**Definition 2** ($C^1$ generalization). *Given $\delta > 0$, there exist $\mathcal{E}_0, \mathcal{E}_1 > 0$ and a function $(\delta, \mathcal{E}_0, \mathcal{E}_1) \to \tau(\delta, \mathcal{E}_0, \mathcal{E}_1) \in \mathbb{N}$ such that $\mathbb{E}_{x \sim \mu}\|F(x) - F_{\mathrm{nn}}(x)\| \leq \mathcal{E}_0$ and $\mathbb{E}_{x \sim \mu}\|dF(x) - dF_{\mathrm{nn}}(x)\| \leq \mathcal{E}_1$ for all $m \geq \tau$ with probability $\geq 1 - \delta$ over the randomness of the training data from $\mu^m$.*

We now make an optimistic assumption on a learned model, $F_{\mathrm{nn}}$, that satisfies the above definition of $C^1$ generalization. Using Hoeffding's inequality, we know that for any $\delta_0 > 0$, with probability at least $1 - \delta_0$ over the randomness of $x$, $\|F(x) - F_{\mathrm{nn}}(x)\| \leq \mathcal{E}_0 + (\sup_{x \in M} \|F(x) - F_{\mathrm{nn}}(x)\|)\sqrt{\log(2/\delta_0)}$ and $\|dF(x) - dF_{\mathrm{nn}}(x)\| \leq \mathcal{E}_1 + (\sup_{x \in M} \|dF(x) - dF_{\mathrm{nn}}(x)\|)\sqrt{\log(2/\delta_0)}$. Given $\delta > 0$, let $\epsilon_0 := 2\mathcal{E}_0$ and $\epsilon_1 := 2\mathcal{E}_1$. Fixing $\delta_0 > 0$, suppose that the trained model $F_{\mathrm{nn}}$ is such that $(\sup_{x \in M} \|F(x) - F_{\mathrm{nn}}(x)\|) < \epsilon_0/(2\sqrt{\log(2/\delta_0)})$ and $(\sup_{x \in M} \|dF(x) - dF_{\mathrm{nn}}(x)\|) \leq \epsilon_1/(2\sqrt{\log(2/\delta_0)})$. Taking a union bound, with probability $> 1 - (\delta + \delta_0)$, $\|F(x) - F_{\mathrm{nn}}(x)\| \leq \epsilon_0$ and $\|dF(x) - dF_{\mathrm{nn}}(x)\| \leq \epsilon_1$. We enhance this inequality to obtain a stronger assumption on $F_{\mathrm{nn}}$.

**Assumption 1** ($C^1$ strong generalization). *Given $\delta > 0$, there exist $\epsilon_0, \epsilon_1 > 0$ and a function $(\epsilon_0, \epsilon_1, m) \to n(\epsilon_0, \epsilon_1, m) \in \mathbb{N}$ such that $\|F(F_{\mathrm{nn}}^t(x)) - F_{\mathrm{nn}}^{t+1}(x)\| \leq \epsilon_0$ and $\|dF(F_{\mathrm{nn}}^t(x)) - dF_{\mathrm{nn}}(F_{\mathrm{nn}}^t(x))\| \leq \epsilon_1$ for all $t \leq n$, $m \geq \tau(\delta, \epsilon)$, and $n \to \infty$ as $m \to \infty$, with probability $\geq 1 - \delta$ over the initial state $x$.*

That is, we assume that, with high probability, the trained model makes a small error at each time. This stronger notion of generalization can be satisfied when the true model shows a smooth linear response in its statistics [Bal14], or in practice, training is performed with points sampled at random near the attractor, as opposed to with a spin-off time to achieve a state on the attractor. Given a tuple, $(\epsilon_0, \epsilon_1)$, an orbit with initial condition $x$ that satisfies, $\|F(F_{\mathrm{nn}}^t(x)) - F_{\mathrm{nn}}^{t+1}(x)\| \leq \epsilon_0$ and $\|dF(F_{\mathrm{nn}}^t(x)) - dF_{\mathrm{nn}}(F_{\mathrm{nn}}^t(x))\| \leq \epsilon_1$ for all $t \leq m$ will be referred to as an $(\epsilon_0, \epsilon_1)$ orbit. That is, at each time, the neural representation $F_{\mathrm{nn}}$ is $(\epsilon_0, \epsilon_1)$-close to the true map, $F$. Under this assumption, we can follow the proof of the Shadowing lemma (see e.g., Ch 18 of [KKH95]) for hyperbolic maps to show that a true orbit (of $F$) shadows every $(\epsilon_0, \epsilon_1)$ orbit.

**Proposition 1** (Shadowing). *Let $F_{\mathrm{nn}}$ be an approximation of $F$ that satisfies the $C^1$ strong generalization (Assumption 1). Given any $\delta > 0$, there exist $\epsilon_0, \epsilon_1, n$ such that every $(\epsilon_0, \epsilon_1)$ orbit is $\delta$-shadowed by an orbit of $F$. That is, there is a true orbit, say, $\{F^t(x)\}$, corresponding to every orbit, $\{G^t(x')\}$ such that $\|F^t(x) - G^t(x')\| \leq \delta$, for all $t \leq n$.*

See section A.1 for the proof. Let $\{x_t^{\mathrm{nn}}\}_{t \leq n}$ be an $n$-length orbit of $F_{\mathrm{nn}}$, i.e., $x_{t+1}^{\mathrm{nn}} = F_{\mathrm{nn}}(x_t^{\mathrm{nn}})$. We use $T_{n,\mathrm{nn}}M$ to denote the direct sum $\oplus_{i=1}^n T_{x_i^{\mathrm{nn}}}M$ of tangent spaces along the orbit, $x_t^{\mathrm{nn}}$. The proof follows Theorem 18.1.3 of [KKH95] to apply contraction mapping on a compact ball in $T_{n,\mathrm{nn}}M$.

The above result defines, for each $(\epsilon_0, \epsilon_1)$-orbit, $\{x_t^{\mathrm{nn}}\}_t$, a shadowing orbit, $x^{\mathrm{sh}} := \{F(x_t^{\mathrm{nn}} + v_t)\}_t$, where $v = \oplus_t v_t$ is the fixed point of the contraction map in the proof (section A.1). Let $\mu_n^{\mathrm{sh}}(x_0^{\mathrm{nn}})$ be the empirical measure defined on the shadowing orbit corresponding to an $(\epsilon_0, \epsilon_1)$-orbit, $\{x_t^{\mathrm{nn}}\}_t$, i.e., $\mu_n^{\mathrm{sh}}(x_0^{\mathrm{nn}}) = \mathrm{Unif}\{x_0^{\mathrm{sh}}, \cdots, x_t^{\mathrm{sh}}, \cdots, x_{n-1}^{\mathrm{sh}}\}$. A shadowing orbit is indeed an orbit of the true map $F$, but, unexpectedly, it may be atypical for the physical measure, $\mu$. That is, for an atypical shadowing orbit, the time average, $(1/n) \sum_{t \leq n} f(x_t^{\mathrm{sh}})$ does not converge to the expected value $\mathbb{E}_{x \sim \mu} f(x)$, as $n \to \infty$. This means that the Wasserstein distance, $W^1(\mu_n^{\mathrm{sh}}(x_0^{\mathrm{nn}}), \mu)$, does not converge to 0 as $n \to \infty$.

Given an initial condition, $x_0^{\mathrm{nn}}$, of an $(\epsilon_0, \epsilon_1)$-orbit, the corresponding shadowing orbit may be typical with some probability (over the distribution of $x_0^{\mathrm{nn}}$), and this probability of finding typical shadowing orbits is a property of the true dynamics, $F$. When $\mu$, the physical measure of $F$, is highly

sensitive to perturbations of $F$, (see [Rue09, Bal14] for surveys on *linear response theory*, the study of perturbations of statistics) this probability may be small. Although the connection between the sensitivity of statistics and the atypicality of shadowing orbits is not completely known, this can justify the differences in the statistical accuracy of neural parameterizations with good $C^1$ generalization (see Definition 2). A neural model $F_{\mathrm{nn}}$ which generalizes well in the $C^1$ sense (Definition 2) can be thought of as a $C^1$-smooth perturbation of the true dynamics $F$. If smooth perturbations of $F$ can cause a large change in $\mu$, then even when the training size $m \to \infty$, and the orbit length $n \to \infty$, $W^1(\mu_n^{\mathrm{sh}}(x_0^{\mathrm{nn}}), \mu)$, may not converge to zero, resulting in a model that does not preserve the *physical* behavior of $F$. On the other hand, for typical shadowing, this possibility is excluded and gives us a characterization of statistically accurate learning.

**Theorem 1** (Statistically accurate learning). *Let $F_{\mathrm{nn}}$ be a model of $F$ that satisfies $C^1$ strong generalization. In addition, let $F_{\mathrm{nn}}$ and $F$ be such that for any $\delta > 0$, there exists an $\epsilon_2 > 0$ so that $\lim_{n \to \infty} W^1(\mu_n^{\mathrm{sh}}(x), \mu) \leq \epsilon_2$ with probability (over the randomness of $x$) $\geq 1 - \delta$. Then, for any $\delta > 0$, there exists an $\epsilon > 0$ such that $\lim_{n \to \infty} W^1(\mathrm{Unif}\{x, F_{\mathrm{nn}}(x), \cdots, F_{\mathrm{nn}}^t(x), \cdots, F_{\mathrm{nn}}^n(x)\}, \mu) \leq \epsilon$ with probability $\geq 1 - \delta$.*

*Proof*: Let $\mu_n^{\mathrm{nn}}(x) := \mathrm{Unif}\{x, F_{\mathrm{nn}}(x), \cdots, F_{\mathrm{nn}}^t(x), \cdots, F_{\mathrm{nn}}^{n-1}(x)\}$ be the empirical measure of an $n$-length orbit of $F_{\mathrm{nn}}$ starting at $x$. Given a $\delta > 0$, we choose $(\epsilon_0, \epsilon_1)$ so that $C^1$ strong generalization is satisfied with probability $\geq 1 - \delta/2$. Thus, an initial condition $x$ is such that $\{F_{\mathrm{nn}}^t(x)\}_{t \leq n}$ is an $(\epsilon_0, \epsilon_1)$ orbit, where the tuple $(\epsilon_0, \epsilon_1)$ is as defined in Proposition 1, with probability $\geq 1 - \delta/2$. Applying Proposition 1, we have, for any 1-Lipschitz function $f : M \to \mathbb{R}$, $(1/n) \sum_{t \leq n} |f(F_{\mathrm{nn}}^t(x)) - f(F^t(x))| \leq (1/n) \sum_{t \leq n} \|F_{\mathrm{nn}}^t(x)) - F^t(x)\| \leq \delta/2$. Taking a supremum over $f$, $W^1(\mu_n^{\mathrm{sh}}(x), \mu_n^{\mathrm{nn}}(x)) \leq \delta/2$, with probability $\geq 1 - \delta/2$. By assumption, there exists some $\epsilon_2 > 0$ such that $\lim_{n \to \infty} W^1(\mu, \mu_n^{\mathrm{sh}}(x)) < \epsilon_2$ with probability $1 - \delta/2$. Hence, $\lim_{n \to \infty} W^1(\mu, \mu_n^{\mathrm{nn}}(x)) \leq \lim_{n \to \infty} W^1(\mu_n^{\mathrm{sh}}(x), \mu_n^{\mathrm{nn}}(x)) + W^1(\mu, \mu_n^{\mathrm{sh}}(x)) \leq \epsilon_2 + \delta/2 := \epsilon$, with probability $> 1 - \delta$, using triangle inequality and taking union bound.

This result explains why training to minimize Jacobian-matching loss (3) can lead to statistically accurate models, even though, long-time temporal patterns in the data are *not* learned explicitly by regression for the one-time map $F$. Since $C^0$ generalization (i.e., small errors in(2)) is insufficient for learning shadowing orbits, and thus for Proposition 1 and Theorem 1 to hold, the models trained on MSE loss 1 are not expected to learn ergodic/statistical averages with respect to $\mu$.

When shadowing orbits are atypical with high probability, we observe numerically that $C^1$ generalization, i.e., training with Jacobian-matching loss, still does not produce statistically accurate dynamics, in line with the above theorem. For instance, for maps with atypical shadowing described in [CW21], we find that learned neural representations with Jacobian-matching do have good $C^1$ generalization (Figure 6), but do not exhibit good statistical accuracy and learn incorrect Lyapunov exponents (see Table 5, Plucked Tent map).

## 4 Dynamic generative models

So far, we have focused on understanding the statistical accuracy of supervised learning of dynamical systems. Without any minimization of distances on the space of probability measures, we proved sufficient conditions under which regression with Jacobian-matching information can yield samples from $\mu$ with high probability. A generative method is an unsupervised learning technique to train on samples from a target distribution to produce more samples (provably) from the target. Naturally, we can use several popular generative models for our target physical measure here, including score-based methods [SE19, SSDK+20], Variational Autoencoders (VAE) [RM15] or normalizing flows [RM15, PNR+21]. However, these methods neglect the dynamical relationships in the input samples. In other words, from a vanilla generative model of a physical measure, we cannot also recover the true dynamics. Thus, we focus here on Latent SDEs models from [LWCD20], which combine neural representations of dynamics with generative models. We reinterpret them as dynamic generative models, and analyze their ability to faithfully represent both the underlying dynamics as well as the physical measure.

In a dynamic generative model, we approximate $F^t$ with a stochastic map, that can written as, $F_{\mathrm{ls}} := f_\theta \circ \Phi_\phi^t \circ g_\phi$, where the subscript ls stands for "latent SDE". Here, the function $g_\phi : \mathbb{R}^d \to \mathbb{R}^{d_l}$, with learnable parameters, $\phi$, is a (possibly stochastic) embedding from the data to latent

space ($\mathbb{R}^{d_l}$), such that, $g_{\phi\sharp}\mu = q_{\phi,0}$. Recall the pushforward notation ($\sharp$), i.e., if $x \sim \mu$, then, $g_\phi(x) \sim q_{\phi,0}$. The dynamical system $\Phi_\phi^t : \mathbb{R}^{d_l} \to \mathbb{R}^{d_l}$ acts on the latent space, and defines a sequence of pushforward distributions $\Phi_{\phi\sharp}^t q_{\phi,0} = q_{\phi,t}$. A special case of this setup is a "latent ODE", where $\Phi_\phi^t$ is a deterministic map instead. With a stochastic latent SDE model instead, we observe, in line with [LWCD20], that the multimodal distribution of the Lorenz '63 attractor is reproduced better. That is, $\Phi_\phi^t$ is a solution map of a Neural SDE: $d\Phi_\phi^t(z) = w_\phi(t, \Phi_\phi^t(z))dt + \sigma_\phi(t, \Phi_\phi^t(z)) \circ dW_t$. Here the drift term, $w_\phi$, and the diffusion term, $\sigma_\phi$, are represented as neural networks. The decoder $f_\theta : \mathbb{R}^{d_l} \to \mathbb{R}^d$ is a deterministic map that defines the conditional, $f_{\theta\sharp}q_{\phi,t} = p_\theta(\cdot|Z_t)$. This dynamic VAE approach has been found to be expressive for chaotic systems (see Chapters 3 and 5 of [Kid22], [KMFL20]), wherein the conditional distribution, $z \to q_{\phi,0}(z|X_{1:m})$, of $Z_0$ given $X_{1:m} := \{F^t(x)\}_{t \le m}$ is modeled as a Gaussian distribution whose learnable parameters are also denoted by $\phi$. Similarly, the conditional $p_\theta(\cdot|Z_t)$ is again modeled as a Gaussian with parameters $\theta$. The parameters, $\phi$ and $\theta$ respectively, of the encoder and the decoder are trained by maximizing the following *dynamic* version of the evidence lower bound (ELBO): $\ell_{ls}(X_{1:m}, (\phi, \theta)) := \sum_{t=1}^m \mathbb{E}_{z_t \sim q_{\phi,t}(\cdot|X_{1:m})}[-\log p_\theta(x_t|z_t)] + \mathrm{KL}(q_{\phi,0}(\cdot|X_{1:m})\|p_{Z_0})]$, where the prior $p_{Z_0}$ follows a standard Gaussian distribution in the latent dimension. For alternatives to the ELBO objective above, such as the Wasserstein-GAN objective, we refer the reader to [Kid22].

We now evaluate both the learned dynamics, $F_{ls}$, and the learned generative model, $p_\theta$, which approximates $\mu$. In Figure 11, we present the empirical distribution of a generated orbit against that of a typical orbit of the Lorenz system ($\mu$). We observe that the distributions match well, nevertheless the vector field is not well-approximated. First, even though the system is deterministic, the learned $\Phi_\phi^t$ with minimum generalization error ($\mathbb{E}\ell^{stat}$) encountered in the hyperparameter search (Appendix C.8) is not, i.e, the diffusion term $\sigma_\phi$ is not zero. The learned stochastic map, $F_{ls}$, produces an incorrect stable LE for the Lorenz system ($\approx -11.8$), while the leading unstable LE matches reasonably well.

Since the map $F_{ls}$, or its underlying vector field, on the latent space is not unique, we may obtain maps that do not preserve dynamical structure or invariants, even if the generated samples from $p_\theta$ approximately capture $\mu$. As noted in section 2, the physical measure $\mu$ is often singular, but absolutely continuous on lower-dimensional manifolds, leading to lack of theoretical guarantees for vanilla generative models [Pid22]. Finally, the sample complexity (and tight generalization bounds) of generative models, the above variational optimization, are not fully understood theoretically, especially for singular distributions (that satisfy the *manifold* hypothesis [BCV13]). We remark that since the minimax rates for approximating distributions have an exponential dependence on the dimension, exploiting the intrinsic dimension (unstable dimension) associated with the support of $\mu$ will be key to tractable generative models for $\mu$ in high-dimensional chaotic systems. Even in the Lorenz '63 system, we require $\mathcal{O}(10^6)$ samples for training reasonably accurate model in Figure 11, while the Jacobian-matching (Figure 1 column 5) produces smaller Wasserstein distances with fewer samples ($10^4$), and a simpler regression problem as opposed to variational optimization above.

## 5 Numerical Experiments

We conduct experiments with the MSE and JAC losses in (1) and (3) respectively on many canonical chaotic systems: 1D tent maps, 2D Bakers map, 3D Lorenz '63 system and the Kuramoto Sivanshinsky equation (127 dimensional system after discretization) (Appendix C). In each system, we identify the Neural ODE model with the lowest generalization errors $R$ and $R_\lambda$ in (2) and (4) respectively, by an architecture and optimization hyperparameter search (Appendix B) . On these models that generalize well, we perform tests of statistical accuracy and LE computations (as described in section 2 for the Lorenz '63 system). Consistent with Theorem 1, although most of the considered systems are not uniformly hyperbolic, we find that the JAC models are statistically accurate and reproduce the LEs, while the MSE models with $R$ comparable to $R_\lambda$ are not statistically accurate. For the KS system for instance, the MSE models even overpredict the number of positive LEs. Interestingly, the best JAC model can learn more than half of the first 64 LEs, compared to the best MSE model that can learn only 2 out of 64 LEs, with $< 10\%$ relative error. The Python code is available at https://github.com/ni-sha-c/stacNODE.

# 6 Related work

**Neural ODEs and generalization.** Introduced as continuous-time analogues [Wei17, HR17, CRBD18] of Residual neural networks (ResNets) [HZRS16], Neural ODEs [CRBD18, HR17] offer a vector field parameterization that can be time-integrated with an ODE solver. Augmented Neural ODEs [DDT19, RCD19, DA22] demonstrate improved expressivity for complex dynamics, and some universal approximation results appear in [ZGUA20, Kid22]. Neural ODEs [CRBD18] as normalizing flows [RM15] and for density estimation have been tackled in [GCB+19]. Training and regularization techniques that allow handling long time series are the subject of numerous works [RMM+20, KJD+21, FJNO20, PER23a, PER23b, PMSR21, GBD+20, MSKF21]. Our focus is different: we characterize when an elementary Jacobian-matching regularization serves as an inductive bias toward learning physical representations from irregular, even chaotic data.

**Data-driven surrogates of complex systems.** The vast and growing literature in the field of physics-informed machine learning[VAUK22, LKB22, PABT+21, HZB+21, vdGSB+20, WY21, RPK19, WY21, KKLL21, LK22] has encouraged the adoption of physical machine learning models that preserve physical properties, symmetries and conservation laws, and are yet applicable in scientific problems [HKUT20, PABT+21, HZB+21, RBL22, VAUK22, vdGSB+20, YHP+23]. At the same time, several purely data-driven methods for complex systems [CNH20, PSH+22], which do not require an expensive high-fidelity solver, have gained attention for their impressive prediction skill [RCVS+19, SLST17, LSGW+23] and both faster training and inference making them suitable for optimization [BB21, RBL22, LP21, JD21] and inverse problems [HVT23, AEOV23]. Since the fundamental regression problem we study underlies both hybrid and data-driven methods, we provide insight into dynamics-aware generalization (e.g., reproducing ergodic behavior, Lyapunov exponents etc) applicable to different surrogate modeling approaches. Several innovative approaches for ensuring training stability [MMD22, SWP+24, HMBD23, JLOW24] in chaotic systems when using recurrent architectures have been proposed recently. The failure of generalization notions based on mean-squared error have also been noted in [SWP+24, JLOW24] and empirical strategies and new definitions of generalization suitable of non-ergodic systems have been introduced in [GHB+24]. For ease of theoretical analysis, we do not consider these more sophisticated training approaches, choosing instead to learn short-term dynamics which obviates the need for stabilization strategies. Encouragingly, we observe low sample complexity of vanilla regression with Jacobian information to learn physical measures, when compared to the generative modeling approaches (which can be comparable to RNNs as well). A computational analysis of the Jacobian loss training compared to generative modeling/stablized recurrent training is deferred to a future work.

**Ergodic theory and shadowing.** Hyperbolic dynamics and ergodic theory (see e.g. the textbook [KKH95]) lay the foundation for understanding the long-time/statistical physical behavior [You02] of complex systems. The scientific computing community has leveraged ergodic theory and shadowing [Ano67, Bow75, Pil06] for rigorous computations that use high-fidelity numerical simulations of chaotic systems [GL24, Wan13, Ni21] and to analyze the correctness of numerical simulations [HYG87, CW21, Lia17, Sau05, GHYS90]. The novelty of our work lies in introducing shadowing as the basis for generalization, thus providing new analysis tools to understand the correctness of learned chaotic systems [LLSK+22]. An interesting direction for future work is to extend dynamics-aware generalization bounds similar to Theorem 1 to operator learning with Sobolev norms introduced in [LLSK+22].

# 7 Conclusion

Our dynamics-aware generalization (Theorem 1) and empirical results provide a new characterization of statistical accuracy in models learned from dynamical data. These results open many avenues for improving mechanistic understanding and bridging the theory-practice gap in physical neural modeling of complex systems:

**Understanding learning attractors.** By exposing foundational problems in the elementary and fairly general setting of regression of a dynamical system, our analytical tools in section 3 broaden our conceptual understanding of learning from time series data in more complicated models and paradigms.

**Learning dynamical representations vs. generative models.** We show that physical measures can be produced by accurate neural representations of the dynamics, which can be more data and time-efficient compared with generative modeling for time series generated by chaotic systems.

**Dynamics-aware learning of scientific models.** Our results imply that generalization does not imply statistical accuracy and preservation of dynamical invariants and properties such as Lyapunov exponents, which is crucial for trustworthy ML for science. Thus, we reinforce the need to move toward a context-aware theory of generalization, organically unifying complex dynamics with learning theory.

**Limitations:** A limitation of the theoretical results about $C^1$ generalization is that we need to assume the typicality of shadowing. Empirically, the Jacobian can be expensive to estimate for high-dimensional scientific applications. We leave the study of statistical accuracy for learning atypical shadowing orbits, and the extension of an efficient algorithm for Jacobian information during training as a future work.

**Acknowledgments** We are grateful to the anonymous reviewers for their useful feedback and their suggestions of additional numerical results, which have strengthened the paper. NC acknowledges support from the James C. Edenfield faculty fellowship provided by the College of Computing at Georgia Tech.

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

# A    Proofs and assumptions

Our results in section 3 make the assumption that the map $F$ is uniformly hyperbolic. Roughly speaking, this means the uniform (on the attractor $\Lambda \subseteq M$) expansion and contraction of infinitesimal perturbations under the differential $dF$. More precisely, in a uniformly hyperbolic system, there exists a decomposition of the tangent bundle, $TM = E^u \oplus E^s$, into an unstable ($E^u$) and stable ($E^s$) subbundle such that these subbundles are $dF$-invariant, and moreover, for any vector field $v \in E^s$, $\|dF^t v\| \leq C \lambda^t \|v\|$, $t \in \mathbb{Z}^+$. Similarly, an infinitesimal perturbation along vector field $v \in E^u$ shows a uniform exponential decay in norm backward in time, $\|dF^t v\| \leq C \lambda^{|t|} \|v\|$, $t \in \mathbb{Z}^-$. Here the constants $C, \lambda$ are uniform over $\Lambda$. There are several relaxations of the uniformity in the splitting to obtain nonuniform and partial hyperbolicity; there are also several known examples of non-hyperbolic chaotic systems. Assumptions of hyperbolicity are common in dynamical systems analyses due to well-understood ergodic theoretic results, including existence of physical measures of the SRB-type [You02, You17]. Such assumptions are also widespread in computational dynamics, wherein algorithms derived rigorously for uniformly hyperbolic systems are found to be applicable [WG18] to turbulent fluids and chaotic systems in practice [GC95, EHL04, CW22], where a rigorous verification [BJS24] of these assumptions is not feasible.

We remark that we only require high-probability finite-time shadowing, and thus, our results can possibly be extended under relaxations of uniform hyperbolicity. Despite being mathematically convenient, the class of uniformly hyperbolic systems do contain examples of the pathological behaviors we address: atypicality of shadowing and large linear responses. Moreover, our numerical examples, including our primary one – the Lorenz '63 system, which is only singular hyperbolic – are chosen to not all be uniformly hyperbolic systems, in order for our inferences to have wider applicability.

## A.1    Proof of Proposition 1

Here, we complete a proof sketch that we begin in section 3. An element $v \in T_{n,\mathrm{nn}}M$ with $v = \oplus_{t \leq n} v_t$ can be identified as $t \to v_t \in T_{x_t^{\mathrm{nn}}}M$. We define a function $\mathcal{F}_{\mathrm{nn}} : T_{n,\mathrm{nn}}M \to T_{n,\mathrm{nn}}M$ as $\mathcal{F}_{\mathrm{nn}}(v)_{t+1} = x_{t+1}^{\mathrm{nn}} - F(x_t^{\mathrm{nn}} + v_t)$. If $v$ is a fixed point of $\mathcal{F}_{\mathrm{nn}}$, that is, $\mathcal{F}_{\mathrm{nn}}(v) = v$, then, $\{x_t^{\mathrm{nn}} + v_t\}_{t \leq n}$ is an orbit of $F$. Writing $\mathcal{F}_{\mathrm{nn}}(v) = \mathcal{F}_{\mathrm{nn}}(0) + d\mathcal{F}_{\mathrm{nn}}(0)v + N_{\mathrm{nn}}(v)$, where $N$ is the nonlinear part of $\mathcal{F}_{\mathrm{nn}}$, a fixed point $v$ satisfies, $(\mathrm{Id} - d\mathcal{F}_{\mathrm{nn}}(0))v = \mathcal{F}_{\mathrm{nn}}(0) + N_{\mathrm{nn}}(v)$. To show that $v$ exists, we follow Theorem 18.1.3 of [KKH95] and prove that $\mathcal{T}_{\mathrm{nn}}(w) = (\mathrm{Id} - d\mathcal{F}_{\mathrm{nn}}(0))^{-1}(N_{\mathrm{nn}}(w) + \mathcal{F}_{\mathrm{nn}}(0))$ is a contraction on a compact subset of $T_{n,\mathrm{nn}}M$. We have $\|\mathcal{T}_{\mathrm{nn}}(v) - \mathcal{T}_{\mathrm{nn}}(w)\| \leq \|(\mathrm{Id} - d\mathcal{F}_{\mathrm{nn}}(0))^{-1}\| \|N_{\mathrm{nn}}(v) - N_{\mathrm{nn}}(w)\| \leq C(\epsilon_0, \epsilon_1)\|N_{\mathrm{nn}}(v) - N_{\mathrm{nn}}(w)\|$ with probability $> 1 - \delta$. To see this, note that $d\mathcal{F}_{\mathrm{nn}}(0)(v) = dF v$, and $F$ is a hyperbolic map, by assumption, and hence Lemma 18.1.4 of [KKH95] applies. Next, assuming that $\sup_{x \in M} \|d^2 F(x)\|$ is bounded (i.e., $dF$ is Lipschitz), we obtain that $dN_{\mathrm{nn}}$ is Lipschitz, and hence, $\|\mathcal{T}_{\mathrm{nn}}(v) - \mathcal{T}_{\mathrm{nn}}(w)\| \leq \|(\mathrm{Id} - d\mathcal{F}_{\mathrm{nn}}(0))^{-1}\| \|N_{\mathrm{nn}}(v) - N_{\mathrm{nn}}(w)\| \leq C(\epsilon_0, \epsilon_1)K\delta_0\|v - w\|$ with probability $> 1 - \delta$. Here, $\max_t\{\|v_t\|, \|w_t\|\} \leq \delta_0$, which is chosen independent of $n, \epsilon_0, \epsilon_1$ and $\delta$, and $K$ is the Lipschitz constant of $dN_{\mathrm{nn}}$. Thus, $\mathcal{T}_{\mathrm{nn}}$ is a contraction on a $\delta_0$ ball around $0 \in T_{n,\mathrm{nn}}M$ when $C(\epsilon)K\delta_0 < 1 - \epsilon_2$, for some $\epsilon_2 > 0$. Since $\|\mathcal{T}_{\mathrm{nn}}(0)\| \leq C(\epsilon_0, \epsilon_1)\epsilon_0$, and for any $v$ in a $\delta_0$ ball around in 0 in $T_{n,\mathrm{nn}}M$, $\|\mathcal{T}_{\mathrm{nn}}(v)\| \leq C(\epsilon_0, \epsilon_1)\epsilon_0 + (1 - \epsilon_2)\delta_0$, when $\epsilon_0, \epsilon_1$ and $\delta_0$ are such that $C(\epsilon_0, \epsilon_1)\epsilon_0 + (1 - \epsilon_2)\delta_0 < \delta_0$, we have that $\mathcal{T}_{\mathrm{nn}}$ maps a $\delta_0$ ball around in 0 in $T_{n,\mathrm{nn}}M$ to itself. Thus, the unique fixed point (from contraction mapping theorem on the $\delta_0$ ball) lies within the ball.

# B    Experimental Details

## B.1    Data

In this paper, the time series data from different chaotic systems are generated by simulating the ODEs and iterated function systems below. The ODEs were numerically integrated using the fourth-order Runge-Kutta solver from the torchdiffeq[3] library [CRBD18], with an absolute and relative error tolerance of $10^{-8}$.

---

[3]https://github.com/rtqichen/torchdiffeq

We split the simulated trajectories into training and test datasets. The first 10,000 data points are used as the training data and the last 8,000 points are the test data. The time step size of the simulation is included in Table 2.

**1D: Tent maps** As our first examples, we choose three types of tent maps defined in [CW21], that are shown to be examples exhibiting atypical shadowing orbits: the tilted map (5), the pinched map (6), and the plucked map (7). We use $n = 3$ and $s = 0.2$ and $s = 0.8$ in the plucked map following [CW21].

$$F(x; s) = \begin{cases} \frac{2}{1+s}x, & x < 1 + s, \\ \frac{2}{1-s}(2 - x), & x \geq 1 + s; \end{cases} \tag{5}$$

$$F(x; s) = \begin{cases} \frac{4x}{1+s+\sqrt{(1+s)^2-4sx}}, & x < 1, \\ \frac{4(2-x)}{1+s+\sqrt{(1+s)^2-4s(2-x)}}, & 2 \leq x \leq 1; \end{cases} \tag{6}$$

$$F_{s,n}(x) = \min(\lambda_{s,n}(x), \lambda_{s,n}(2 - x)), \qquad 0 < x < 2, \tag{7}$$

where

$$f_s(x) = \min(\frac{2x}{1-s}, 2 - \frac{2(1-x)}{1+s}), \qquad x < 1,$$

$$o_s(x) = \begin{cases} \frac{f_s(2x)}{2}, & x < 0.5, \\ 2 - \frac{f_s(2-2x)}{2}, & x \geq 0.5, \end{cases}$$

$$\lambda_{s,n}(x) = \frac{o_s(2^n x - \lfloor 2^n x \rfloor)}{2^n} + 2\frac{\lfloor 2^n x \rfloor}{2^n}.$$

**2D: Baker's map** We use the following perturbation of the classical Baker's map from [CW22].

$$F([x, y]^T; s) = \begin{bmatrix} 2x - \lfloor y/\pi \rfloor 2\pi \\ \dfrac{y + s \sin x \sin(2y) + \lfloor x/\pi \rfloor 2\pi}{2} \end{bmatrix} \mod 2\pi.$$

**3D: Lorenz '63** We conduct extensive numerical experiments in this paper with the Lorenz '63 system [Lor63], with $\sigma = 10$, $\beta = 8/3$, and $\rho = 28$, which we describe in section 2.

**3D: Rössler** We use the parameter setting $a = 0.2$, $b = 0.2$, and $c = 5.7$ in the Rössler system [Rös76] below, which is in the chaotic regime.

$$\frac{d\varphi^t}{dt}([x, y, z]^\top) = \begin{bmatrix} -y-z \\ x+ay \\ b+z(x-c). \end{bmatrix}$$

**4D: Hyperchaos** Another test case we consider is the hyperchaotic system below in 4 dimensions from [Zha17], where the parameter values are set as $a = 16, b = 40, c = 20, d = 8$ for the system to show chaotic behavior.

$$\frac{d\varphi^t}{dt}([x, y, z, w]^\top) = \begin{bmatrix} ax + dz - yz \\ xz - by \\ c(x - z) + xy \\ c(y - w) + xz. \end{bmatrix}$$

**127D: Kuramoto-Sivashinsky** To test Neural ODE's performance in learning high dimensional chaotic system, we generate the modified Kuramoto-Sivashinsky (KS) system's solution defined below with a second order finite difference scheme used in [BW14].

$$\frac{\partial u}{\partial t} = -(u+c)\frac{\partial u}{\partial \text{x}} - \frac{\partial^2 u}{\partial \text{x}^2} - \frac{\partial^4 u}{\partial \text{x}^4}$$

$$\text{x} \in [0, L], \quad t \in [0, \infty)$$

$$u(0, t) = u(L, t) = 0$$

$$\left.\frac{\partial u}{\partial \text{x}}\right|_{\text{x}=0} = \left.\frac{\partial u}{\partial \text{x}}\right|_{\text{x}=L} = 0$$

$$u(\text{x}, 0) = u_0(\text{x})$$

## B.2  Architecture

We try two neural network architectures for learning the vector field of the systems: a simple Multi-Layer Perceptron (MLP) model and a ResNet [HZRS16]. For the MLP model, we use GELU[4] as the activation function, and ReLU for ResNet model. In addition, we experiment with adding Fourier layers [LKA+20] to represent the solution operator. We use the Latent SDE code from [LWCD20] for results in section 4.

Table 2: Hyperparameter choices

| Chaotic Systems | Epochs | Time step | Hidden layer width | Layers | Train, Test size | Neural Network | $\lambda$ in (3) |
|---|---|---|---|---|---|---|---|
| Tent map | 10000 | N.A. | [256] | 2 | [10000, 8000] | ResNet | 500 |
| Baker map | 10000 | N.A. | [512] | 3 | [5000, 5000] | ResNet | 100 |
| Lorenz '63 | 8000 | 0.01 | [512] | 7 | [10000, 8000] | ResNet | 500 |
| Rössler | 10000 | 0.01 | [512] | 3 | [10000, 8000] | ResNet | 500 |
| Hyperchaos | 20000 | 0.001 | [512] | 3 | [10000, 8000] | ResNet | 500 |
| Kuramoto-Sivashinsky | 3000 | 0.25 | [512, 256] | 3 | [3000, 3000] | MLP | 1 |

## B.3  Hyperparameter Search

When $v_F(x)$ is a true vector field and $v_h(x)$ is a learned vector field by a neural network, $h$, given solution $x$, relative error can be defined as:

$$\text{relative error}(x) = \frac{\|v_F(x) - v_h(x)\|}{\|v_F(x)\|} \tag{8}$$

Using a grid search, hyperparameter values that yield the lowest relative error in the vector field as per (8) were chosen. Hyperparameter search results are shown in Tables 7, 8, and 9. Hyperparameter values that are different for each system are in the Table 2. We use the AdamW [LH17] optimization algorithm implemented in the PyTorch library for all our experiments.

In addition to MLP and Resnet, for Lorenz '63, we train with FNOs [LKA+20] and latent SDE [LWCD20]. In the FNO network, we fix the number of modes to 4, with a batch size of 200 with 4 Fourier Neural layers. Further details on latent SDE are discussed in section C.8.

## B.4  Computing

Numerical experiments were conducted using Tesla A100 GPUs with 80GB and 40GB memory capacities. All experiments were completed in under one hour, with the exception of those involving the KS system.

---

[4]https://pytorch.org/docs/stable/generated/torch.nn.GELU.html

# C  Additional numerical results

In this section, we present results that are described in section 2 for testing the statistical accuracy and generalization of learned models of the Lorenz '63 system. We also present additional results for the tent maps and the KS equation, described in the previous section.

## C.1  Loss

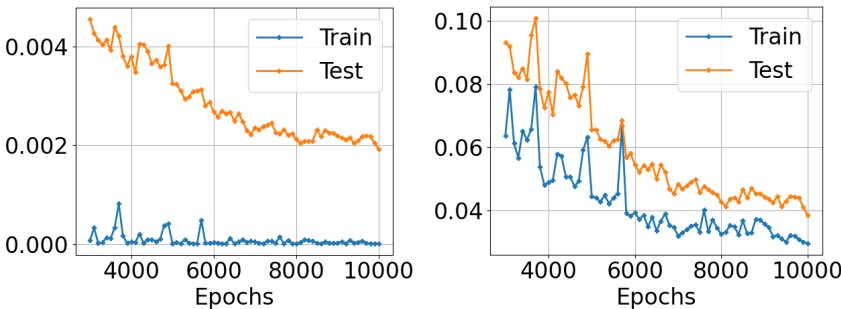

Figure 3: Training and test loss of a representative Neural ODE trained with MSE (1) and Jacobian-matching loss (3).

## C.2  Relative Error

Figures 4 and 5 illustrate the comparison of relative error trends, as defined in Eq. (8), for vector fields learned by a Neural ODE (ResNet) trained with mean squared loss and Jacobian-matching loss (as defined in Eq. (3)), for two distinct dynamical systems.

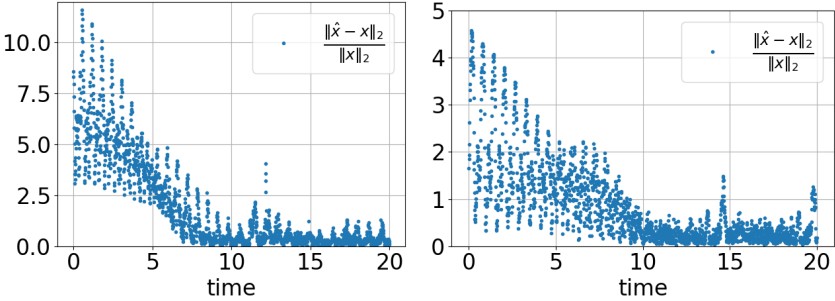

Figure 4: Comparison of relative errors in the vector fields of the Lorenz '63 system produced by a Neural ODE with a ResNet, trained using mean squared loss (Left), and Jacobian-matching loss (Right) as defined in (1) and (3) respectively. The vector field is evaluated on a random true orbit.

## C.3  Tent map

In Figure 6, we show learned models of representative tent map perturbations from the preceding section. We observe that the Jacobian-matching loss leads to reasonably accurate representations of the maps. Yet, the LE computed along learned orbits differ significantly at $s = 0.8$, as shown in Table 5. For various other perturbed tent maps, we show the computed LEs alongside the true LEs in Table 5; for most of the maps, the Jacobian-matching leads to accurate LE predictions, consistent with the results of Theorem 1.

## C.4  Lorenz '63

Here we show identical results to Figure 1 but with ResNet-based Neural ODE models as opposed to MLPs used in Figure 1. We find that Jacobian-matching training leads to superior performance in

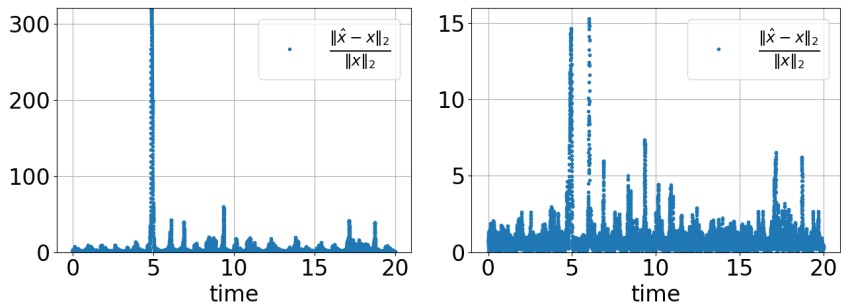

Figure 5: Comparison of relative errors in the vector fields of the Rössler system produced by a Neural ODE with a ResNet, trained using mean squared loss (Left), and Jacobian-matching loss (Right) as defined in (1) and (3) respectively. The vector field is evaluated on a random true orbit.

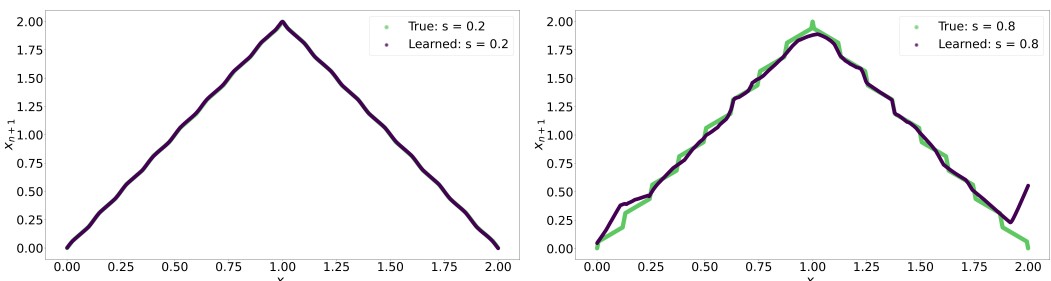

Figure 6: Comparison of true plucked tent map with generated one with Neural ODE trained with Jacobian-matching loss defined in Equation (3). When $s = 0.8$, we observe a failure mode of training with Jacobian-matching loss, possibly due to atypicality of shadowing orbits observed in [CW21]. Details on hyperparameter setting and other statistical results can be found in Table 2 and 5.

terms of distributional match with $\mu$, when compared to MLPs. Furthermore, our overall inference about MSE models leading to atypical orbits still holds, agnostic to modeling and architectural choices.

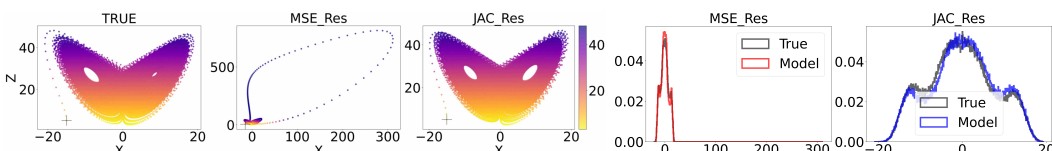

Figure 7: The first 3 columns show orbits on x-z plane obtained from RK4 integration of the Lorenz vector field ([Lor63], the Neural ODE, 'MSE_Res', trained with mean-square loss, and the Neural ODE, 'JAC_Res', trained with Jacobian-regularized loss, respectively (see section 2). The last two columns show the probability distribution of orbits generated by the true (gray), 'MSE_Res' (red) and 'JAC_Res' (blue) models. Experimental settings are in Appendix B.

## C.5 KS equation

After spatial discretization and following the scheme of [BW14], we obtain a 127-dimensional dynamical system representing the KS equation, which we consider to be the ground truth map $F$. Figure 8 shows the solutions of the KS system over physical space x and time (T). Since the original model and the learned models are all chaotic, we expect two solutions of even slightly different models to diverge along the time axis, even when starting with identical initial conditions. We observe that this divergence is minimal for the JAC model, while the errors in the MSE model grow strikingly quickly.

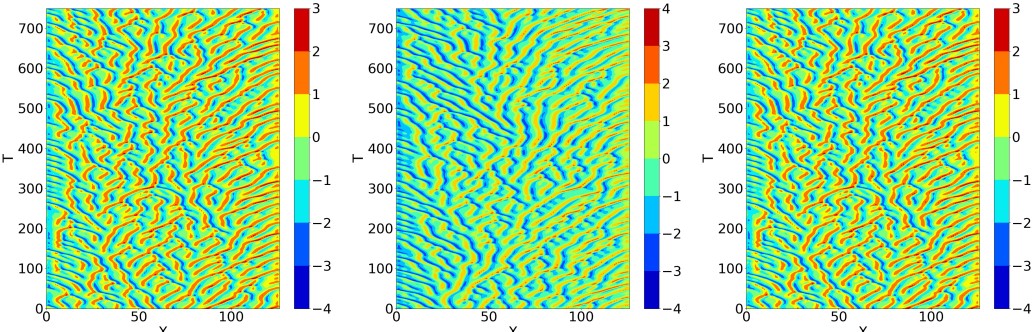

Figure 8: Solution plot of Kuramoto-Sivashinksy system when number of inner nodes is 127 and c=0.4 (see [BW14] for the parameter c). True solution (left), solution of the Neural ODE with mean squared loss (1) (center column), solution of the Neural ODE trained with Jacobian-matching loss defined in (3) (right)

## C.6    Learning Lyapunov exponents

We report Lyapunov exponents learned with Neural ODE trained with mean squared loss and Jacobian-matching loss in Table 5. To compute Lyapunov exponents, we use a QR algorithm of Ginelli et al [GCLP13]. We find that, for most systems considered, the LEs computed by JAC models match very well with the ground truth, while the MSE models sometimes capture the leading LEs but are inaccurate in the rest of them.

## C.7    Comparison of Jacobian-matching loss training with unrolling dynamics

In this section, we report experiments on the Lorenz '63 system with an unrolled loss function:

$$\ell_{\mathrm{u}}(x) := \frac{1}{k} \sum_{t \le k} |v_t(x)|^2, \text{where } v_t(x) = F_{\mathrm{nn}}^t(x) - F^t(x). \tag{9}$$

With $k = 10$ timesteps of unrolling time, we see that the attractor is reproduced well, as shown in Figure 9. However, atypical orbits are still produced for random initializations (Figure 10).

We experiment using two types of $v_{\mathrm{nn}}$, MLP and Resnet, and varying sequence lengths, $k$. As shown in Table 3, we observe that the learned negative Lyapunov exponent plateaued for $k \ge 40$, remaining between -9 and -10 (the true value being $\sim -14.5$), with no further improvement. Also, as $k$ increases, we observe that Neural ODEs overestimate the positive Lyapunov exponent. Overall, unrolling seems to learn more accurate representations than the MSE model but less accurate representations than the JAC models. We also observe that the unrolling time needs to be fine-tuned as a hyperparameter to achieve good generalization; a small perturbation can lead to training instabilities.

To understand these results, for short times, when compared to the Lyapunov time, $v_t$ are in tangent spaces along the orbit. This yields the recursive relationship, $v_t \approx dF(x_{t-1})v_{t-1}$, when $\mathcal{O}(\|v_t\|^2)$ is negligible. Thus, the unrolled loss does contain Jacobian information implicitly although it does not enforce the learned trajectory to be close to the true trajectory in $\mathcal{C}^1$-distance. We remark, speculatively, that Theorem 2 gives a possible explanation for why the unrolling loss performs better than a one-step loss (even in some practical climate emulators, e.g., FourcastNet [PSH+22]) at learning the physical measure. In other words, our numerical results lend support to the central thesis of this paper: adding Jacobian information improves statistical accuracy. Table 4 shows the norm difference between the reproduced invariant statistics and the true statistics of Lorenz '63.

## C.8    Latent SDE: experimental details

Here we present the experimental details and results of learning the Lorenz '63 system with latent SDEs [LWCD20] that are described in Section 4.

As training data, we use 1024 time series of the interval $[0, 6]$ and timestep size 0.01. With a smaller sample size, we empirically observe a large distributional mismatch (see Figure 11). We also observe

Table 3: Lyapunov Spectra learned by Neural ODE models trained on the MSE (1) for multi-step prediction.

| | $v_{nn}$ | |
| $k$ | MLP | ResNet |
|---|---|---|
| 10 | [0.89, -0.0006, -5.53] | [0.77, -0.0080, -4.48] |
| 20 | [0.89, 0.0349, -6.31] | [0.82, 0.0128, -5.19] |
| 30 | [0.906, 0.0018, -7.18] | [0.89, -0.0372, -8.02] |
| 40 | [0.96, -0.0587, -9.87] | [0.92, -0.0655, -9.81] |
| 50 | [0.96, -0.0922, -9.88] | [0.89, -0.0532, -10.76] |

Table 4: True vs. learned Lorenz '63 system: comparison of statistics. ($W^1$: Wasserstein-1 Distance, $\Lambda = [\lambda_1, \lambda_2, \lambda_3]^\top$, set of LEs, $\hat{\mu}_T$: empirical distribution of an orbit of length $T$. The subscript NN indicates quantities computed using NN models. The variable $k$ refers to the sequence length used for training.

| | | Norm Difference | | |
| Model | $k$ | $W^1(\hat{\mu}_{500}, \mu_{\mathrm{NN},500})$ | $\|\Lambda - \Lambda_{\mathrm{NN}}\|$ | $|\langle x \rangle_{500} - \langle x \rangle_{500,\mathrm{NN}}|$ |
|---|---|---|---|---|
| MLP | 50 | 1.4184 | 4.4824 | 1.3845 |
| ResNet | 50 | 0.5091 | 2.9456 | 0.2408 |

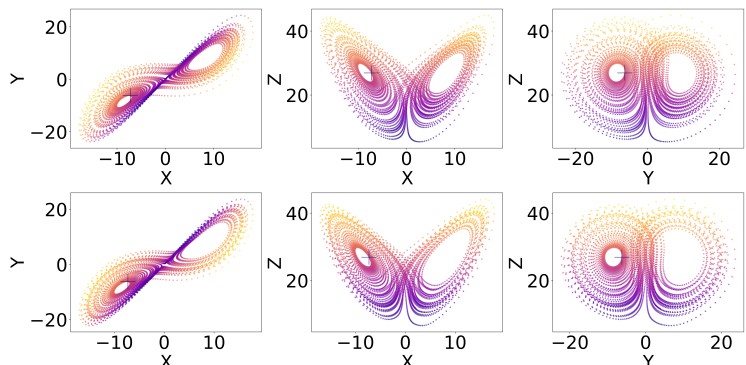

Figure 9: Comparison of the true phase plots of Lorenz '63 with phase plots of Neural ODE trained with the unrolled loss function (9). Initial condition of the long orbits is at $[-9.1164, -3.3816, 33.7482]$. The first row shows the orbits on the xy, xz, and the yz plane obtained from RK4 integration of the Lorenz '63 system. The second row shows the orbits on the xy, xz, and the yz plane generated from Neural ODE trained with the loss (9) with $k = 50$.

that increasing or decreasing the time length of the trajectories can lead to unstable training. As in the supervised learning methods, the time series are simulated using a fourth-order Runge-Kutta scheme (RK4). We follow the same architecture [5] as used in [LWCD20], with a GRU encoder, a linear decoder, and the drift and diffusion functions in the prior and posterior processes modelled by MLPs.

We present the learned Lyapunov exponents in Table 6, and a comparison of the learned and true empirical measures in Figure 11. When learning a deterministic system with a latent SDE, the diffusion coefficient of the learned system is small but not exactly 0, and this results in a slight difference in the Lyapunov exponents ([GHL20]) compared to only using the learned drift term. The latent SDE model was also tested on the stochastic Lorenz attractor [CZH21], where it reproduced the 'bimodal' distribution of the trajectories.

---

[5]`https://github.com/google-research/torchsde/blob/master/examples/latent_sde_lorenz.py`

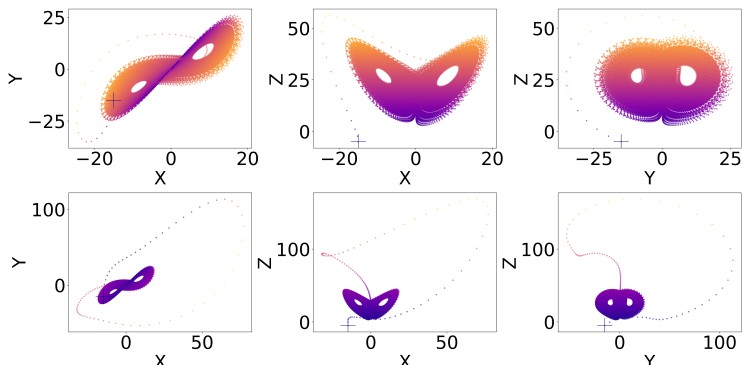

Figure 10: Comparison of the true phase plots of Lorenz '63 with phase plots of Neural ODE trained with the unrolled loss function (9). Initial condition of the long orbits is at $[-15, -15, 5]$. The first row shows the orbits on the xy, xz, and the yz plane obtained from RK4 integration of the Lorenz '63 system. The second row shows the orbits on the xy, xz, and the yz plane generated from Neural ODE trained with the loss (9) with $k = 50$.

Table 5: Chaotic systems and the true Lyapunov Spectra, the learned Lyapunov Spectra from Neural ODEs with MSE loss (1), and the learned Lyapunov Spectra from Neural ODEs with Jacobian-matching loss (3).

| | Lyapunov Spectrum | | |
|---|---|---|---|
| Chaotic Systems | $\Lambda_{\text{True}}$ | $\Lambda_{\text{MSE}}$ | $\Lambda_{\text{JAC}}$ |
| Tent map (tilted, s=0.2) | [0.3188] | [0.6722] | **[0.3402]** |
| Tent map (pinched, s=0.2) | [0.6849] | **[0.6840]** | [0.6836] |
| Tent map (plucked, s=0.2) | [0.6681] | **[0.6763]** | [0.6348] |
| Tent map (tilted, s=0.8) | [0.3188] | **[0.3176]** | [0.3402] |
| Tent map (pinched, s=0.8) | [0.6215] | [0.5770] | **[0.6420]** |
| Tent map (plucked, s=0.8) | [0.3199] | [0.5819] | **[0.5013]** |
| Lorenz '63 | [0.9, 0, -14.52] | [0.87, 0.0091, -4.82] | **[0.88, -0.0012, -14.54]** |
| Rössler | [0.0665, -0.0004 -5.4112] | [0.0008,-0.0285 -1.4108] | **[0.0609, -0.0004 -5.3808]** |
| Hyperchaos | [4.0039, 0.0082 -19.9972, -48.0205] | [4.1393, 0.0955 -15.2120, -29.9480] | **[4.3789, -0.1617 -19.9974, -48.0205]** |
| Kuramoto-Sivashinsky | [ 0.3036, 0.2733, 0.2592, 0.2257, 0.2050, 0.1888, 0.1649, 0.1496, 0.1288, 0.1128, 0.0992, 0.0776, 0.0646, 0.0492, 0.0342 ] | [ 0.1652, 0.1647, 0.1540, 0.1524, 0.1443, 0.1411, 0.1336, 0.1262, 0.1236, 0.1143, 0.1141, 0.1091, 0.1045, 0.0971, 0.0985 ] | **[ 0.2904, 0.2622, 0.2293, 0.1990, 0.1701, 0.1584, 0.1320, 0.1071, 0.0912, 0.0724, 0.0591, 0.0442, 0.0306, 0.0157, 0.0023, ]** |

## C.9 Experimental details of the learned Lorenz '63 models

Table 6: Lyapunov spectra computed from the learned latent SDE model evaluated on Euler-Maruyama and RK4 solvers. Both the learned drift and the near-zero diffusion vector fields are used in Euler-Maruyama, and only the drift vector field is used in RK4.

| Solver | Lyapunov Spectrum |
|---|---|
| Euler-Maruyama | [0.841, 0.0125, -11.88] |
| RK4 | [ -0.44, -0.508, -11.75] |

Table 7: Results of search over hyperparameters (batch size, weight decay, hidden layer depth and width) in training Neural ODEs with MLPs (with fully connected and convolution layers). We train with mean squared loss using the AdamW optimization algorithm, with two values of weight decay: $10^{-3}$ and $10^{-4}$, and an adaptive learning rate with an initial value of 0.001. For each hyperparameter combination, we show the test loss and the relative error in the one-timestep predictions averaged over 8000 samples; we choose the hyperparameter combination that results in the least relative error. The time step of the maps (both true and NNs) are set at 0.01.

| Layer | Hidden Unit | Batch Size | | | | | |
|---|---|---|---|---|---|---|---|
| | | Full | | 1000 | | 2000 | |
| | | $10^{-3}$ | $10^{-4}$ | $10^{-3}$ | $10^{-4}$ | $10^{-3}$ | $10^{-4}$ |
| 3 | 256 | 3.0577, 2.02% | 0.1756, 2.16% | 1.0858, 9.16% | 0.0700, 4.92% | 0.0300, 8.45% | 0.0713, 6.17% |
| | 512 | 0.0967, 2.11% | 0.0637, 1.84% | 0.0262, 7.71% | 3.5441, 6.89% | 0.2689, 5.89% | 0.2144, 3.98% |
| | 1024 | 0.8875, 1.32% | 16.9632 ,3.01% | 0.0217, 7.89% | 0.2425, 3.12% | 0.0300, 8.45% | 0.0488, 11.22% |
| 5 | 256 | 0.0232, 1.70% | 8.2952, 1.12% | 0.7407, 2.48% | 0.2120, 9.03% | 0.0700, 4.92% | 3.9985, 10.19% |
| | 512 | 0.0936, 2.22% | 6.7319, 1.29% | 243.8366, 15.01% | 24.6090, 17.84% | 3.2698, 17.54% | 0.1982, 7.44% |
| | 1024 | 158.9892, 1.03% | 0.0234, 2.03% | 0.0472, 4.94% | 0.0713, 4.02% | 0.2425, 3.12% | 0.0332, 11.46% |
| 7 | 256 | 0.0442, 1.13% | 0.0583, 1.65% | 0.8257, 8.72% | 28.2333, 14.39% | 0.1761, 6.32% | 0.3564, 4.62% |
| | 512 | 0.5062, **1.03%** | 0.0204, 1.41% | 4.5527, 11.59% | 0.0470, 12.36% | 170.5588, 8.51% | 0.0700, 11.94% |
| | 1024 | 162.6586, 1.87% | 0.0502, 1.15% | 0.0302, 7.71% | 3.7282, 19.24% | 117.4206, 1.20% | 0.0502, 1.15% |

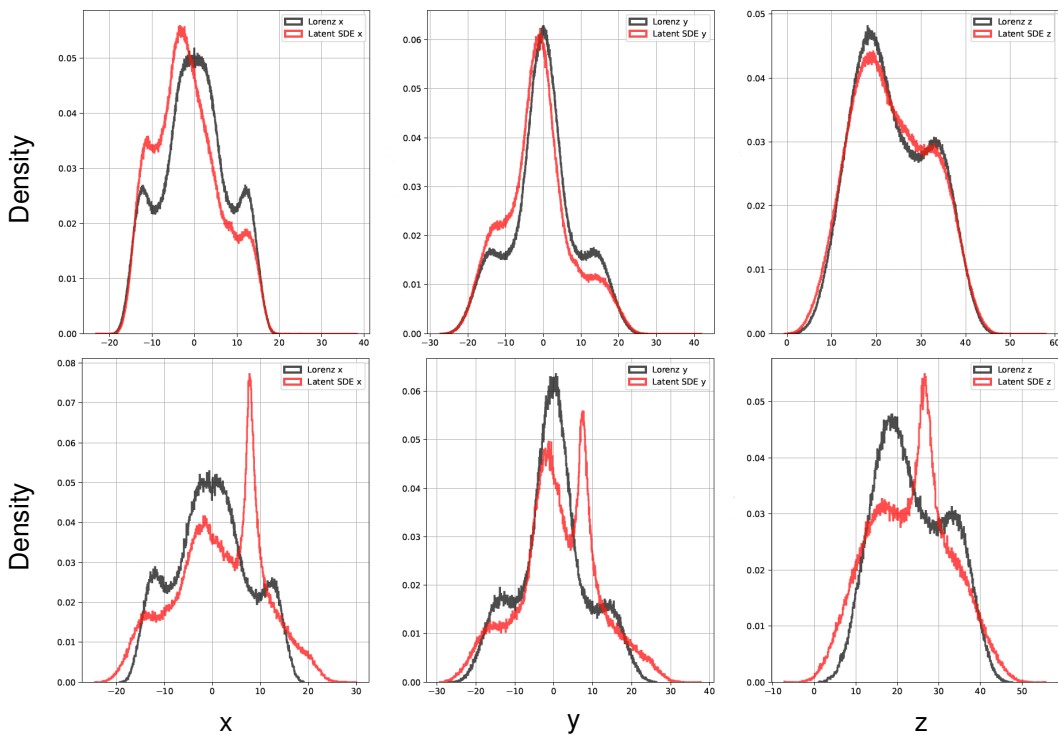

Figure 11: Comparison of empirical distributions of the x, y, and z coordinates of the true orbits of the Lorenz '63 system (black) against those of the latent SDE (red). Top row: The latent SDE model (see section 4) is trained with 614,400 sample points. Bottom row: The latent SDE model is trained with the same sample size, 10,000, as the JAC_MLP and MSE_MLP (Neural ODE) models. We generate a trajectory over the interval $[0, 50]$ for both the Lorenz '63 model and the learned latent SDE system. Gist: we empirically observe that, even when training with $\mathcal{O}(100)$, the latent SDE model results in a worse prediction of the physical distribution compared to supervised learning based on Jacobian-matching loss (3).

Table 8: Results of search over hyperparameters (batch size, weight decay, hidden layer depth and width) in training Neural ODEs with MLP_skip (ResNets). We train with mean squared loss using the AdamW optimization algorithm, with two values of weight decay: $10^{-3}$ and $10^{-4}$, and an adaptive learning rate with an initial value of 0.001. For each hyperparameter combination, we show the test loss and the relative error in the one-timestep predictions averaged over 8000 samples; we choose the hyperparameter combination that results in the least relative error. The time step of the maps (both true and NNs) are set at 0.01.

| Layer | Hidden Unit | Batch Size | | | | | |
| --- | --- | --- | --- | --- | --- | --- | --- |
| | | Full | | 1000 | | 2000 | |
| | | $10^{-3}$ | $10^{-4}$ | $10^{-3}$ | $10^{-4}$ | $10^{-3}$ | $10^{-4}$ |
| 3 | 256 | 0.5759, 2.60% | 0.1186, 3.39% | 0.4406, 9.60% | 0.2987, 6.64% | 0.1015, 4.07% | 0.1698, 5.41% |
| | 512 | 0.1550, 1.80% | 0.0914, ,2.08% | 0.4885, 5.57% | 0.1510, 4.47% | 0.4082, 2.86% | 0.3319, 6.84% |
| | 1024 | 0.4721, 2.23% | 0.1177, 1.76% | 0.0433, 2.11% | 0.0668, 4.07% | 0.1681, 5.15% | 0.2515, 4.89% |
| 5 | 256 | 0.5392, 2.14% | 0.1348, 2.21% | 0.1694, 3.39% | 0.2117, 7.68% | 0.1862, 3.89% | 0.1568, 6.37% |
| | 512 | 0.0437, 1.70% | 0.0611, **1.31%** | 0.2823, 4.87% | 0.1005, 4.74% | 0.0934, 5.97% | 0.0822, 5.23% |
| | 1024 | 0.1914, 2.44% | 0.2093, 2.08% | 0.2429, 8.11% | 0.0563, 1.94% | 0.4757, 2.60% | 0.0377, 4.36% |
| 7 | 256 | 0.1701, 1.91% | 0.1523, 1.99% | 0.8257, 8.72% | 0.9819, 7.22% | 0.1255, 9.65% | 0.3181, 6.80% |
| | 512 | 0.2088 ,2.38% | 0.0806, 1.87% | 0.1857, 4.64% | 0.3120, 5.44% | 0.1538, 4.62% | 0.4729, 7.19% |
| | 1024 | 0.0566, 2.59% | 0.0237, 2.16% | 0.6599, 8.04% | 0.0326, 5.07% | 0.1748, 4.07% | 0.3441, 5.09% |

Table 9: Results of search over hyperparameters (batch size, hidden layer depth and width) in training Neural ODEs with MLP_skip (ResNets). We train with the Jacobian-matching loss (3) using the AdamW optimization algorithm, with a weight decay of $5 \times 10^{-4}$, and an adaptive learning rate with an initial value of 0.001. For each hyperparameter combination, we show the test loss and the relative error in the one-timestep predictions averaged over 8000 samples; we choose the hyperparameter combination that results in the least relative error. The time step of the maps (both true and NNs) are set at 0.01.

| Layer | Hidden Unit | Neural Architecture | | | |
| --- | --- | --- | --- | --- | --- |
| | | MLP | | MLP_skip | |
| | | 500 | 1000 | 500 | 1000 |
| 3 | 256 | 0.0535, 0.22% | 0.0031, 0.26% | 0.3349, 0.96% | 0.3689, 1.52% |
| | 512 | 0.0022, 0.26% | 0.0256, 0.22% | 0.0780, 0.93% | 0.8011, 1.83% |
| | 1024 | 0.3905, 0.44% | 0.0237, 0.82% | 0.3349, 0.96% | 0.2787, 1.80% |
| 5 | 256 | 0.0060, 0.38% | 1.9023, 0.28% | 0.4635, 0.98% | 0.5906, 2.30% |
| | 512 | 0.0300, 0.33% | 0.0612, 0.59% | 0.4211, 0.76% | 0.2773, 2.30% |
| | 1024 | 0.0885, 0.83% | 0.0232, 0.75% | 0.1064, **0.69%** | 0.7675, 2.23% |
| 7 | 256 | 0.0297, 0.20% | 2.8038, 0.48% | 0.1032, 1.01% | 0.5467, 2.60% |
| | 512 | 0.0437, **0.10%** | 0.011, 0.45% | 0.0991, 0.93% | 0.0943, 0.92% |
| | 1024 | 0.4413, 1.3% | 1.2202, 0.36% | 1.1442, 1.24% | 0.1917, 1.43% |

