# OpenReview forum: "When are dynamical systems learned from time series data statistically accurate?"
_NeurIPS.cc/2024/Conference — NeurIPS 2024 poster_

### Official Review · Reviewer_PvwR · 2024-06-26

**Soundness:** 3
**Presentation:** 4
**Contribution:** 2
**Rating:** 6
**Confidence:** 3

**Summary:**

The present manuscript concerns the use of neural networks to fit physical dynamical systems of generic kind (including and focusing on chaotic maps). It is shown that adding the information of the Jacobian of the dynamical map in the supervised learning process leads to better performances.

**Strengths:**

Originality: the work builds on previous ideas and problems. The original part (as claimed by the authors) concerns adding the information on the Jacobian of the dynamical map to the supervised learning process. The author show that this significantly improves the statistical properties of the points generated by the fitted dynamical systems.
Quality: The paper is very well written, well structured and clear. Section 4 is very clear.
Significance: I believe that the problem tackled in this work is a relevant one and the results show an interesting perspective on it.

**Weaknesses:**

I believe that the most important weakness of the manuscript relies on the limitations of the results.
The paper clearly shows that having information on the Jacobian of a dynamical map is essential for reproducing the statistical properties of the generated orbits. However such information is typically lacking.

**Questions:**

1) I believe that it must be carfully stated that it is assumed that the map F does not depend on time. This is an important assumption which may be relaxed.

**Limitations:**

The main and severe limitation is that the improvement training method needs the knowledge of the Jacobian of the ground truth dynamical system. It is unclear how this can be estimated from data contain just points in the orbits of the maps and what are realistic settings where this is known.
I believe that this is a severe limitation of the work. This does not spoil the result in the sense that the manuscript clearly shows that the information coming from the Jacobian is essential, but on the other hand it seems to me that this information is hard to get.

---

> ### Author Rebuttal · Authors · 2024-08-06
>
> ### Results require full Jacobian but it can be approximated with available derivative information in practice
>
> You are absolutely right in that the Jacobian is computationally hard to calculate (through AD or finite difference) in high dimensions; thank you and Reviewer vAp3 for pointing this out! This in turn would slow the training process and might make it prohibitive for learning high-dimensional chaotic systems. We emphasize that our primary contribution is toward the foundational question of understanding generalization in the context of learning chaotic dynamics. Our main result deconstructs the effect of the ergodic properties of the true system on the generalization ability of the neural network. To isolate this effect, we consider only a minimal illustrative setting of regression using the common MSE loss. When some Jacobian information (local information about the short-term dynamics and its linear perturbation structure) is added to the loss, surprisingly we find that the physical measure (global statistics) is learned. Our results are aimed at explaining this observation via the mechanism of shadowing. As a minimal setting to verify our results in practice without any confounders, we consider systems in which the full Jacobian can be calculated and used in training. Having said that however, our results lend themselves to many practical training strategies where the Jacobian information is only approximated. For instance, in scenarios where adjoints or other surrogate models for the Jacobian are available, which are common in many scientific applications in the geosciences and aerospace turbulent flows, these can be used as proxies for the full Jacobian matrix. As we mention in response to Reviewer vAp3, who raised the same concern, we are interested in analyzing whether random Jacobian matrix-vector products can yield informative generalization bounds such as Theorem 2. Since the novelty and focus of our present paper is in the theoretical connection between using Jacobian information and statistical accuracy and why this obviates the need for performing generative modeling, we defer such analyses to future work.
>
> ### Focus on autonomous systems
>
> In this paper, we focus on autonomous systems -- wherein the flow map itself does not depend on time -- which occupy a large class of chaotic systems encountered in nature. As you correctly point out, the analysis and even the background, starting from the existence of a unique physical measure, will completely change when we consider nonautonomous systems such as chaotic systems with control terms and random dynamical systems with chaotic orbits. For these systems, there is a lot of exciting recent development in the theoretical dynamical systems literature (see e.g. this survey article: https://link.springer.com/article/10.1007/s10955-016-1639-0) proving the existence of random physical measures. The next step would be connect the theory of such measures with statistical learning theory. In the present work, we show a path for such a connection in deterministic chaotic systems, deferring random chaotic systems (which are increasingly recognized as being useful models for stochastically parameterized climate systems) to future work. We will add this remark to the appendix in the next revision. Thank you very much for bringing this to our attention!

---

### Official Review · Reviewer_vAp3 · 2024-07-12

**Soundness:** 3
**Presentation:** 2
**Contribution:** 2
**Rating:** 5
**Confidence:** 4

**Summary:**

The paper addresses the problem that classical ERM training of dynamical systems models often fails to capture invariant measures of the observed dynamics, even when test errors are low. The authors use ergodic theory to explain this failure from a theoretical viewpoint. They further demonstrate that incorporating Jacobian information during training enables much better reconstructions of invariant properties and validate their approach on various neural flow operators trained on several common benchmark chaotic systems.

**Strengths:**

- I think the authors study an interesting problem, i.e. when classical ERM for dynamical models can still lead to a reconstruction of invariant properties of the observed dynamical system, as classical ERM (i.e. here simply “one-step-ahead” predictions) comes with benefits of decreased training time and less training difficulties
- The authors use dynamical systems and ergodic theory to accompany and explain the practical behavior of several neural flow operators (N-ODEs, RNNs, MLPs, etc.) when training them using ERM on observed dynamical systems
- The paper is generally well written

**Weaknesses:**

- l. 53 - 56: Indeed there are also connections of training recurrent architectures to model chaotic DS and exploding loss gradients in gradient descent based training [1]. Moreover, recently methods for dealing with these training instabilities have been introduced and proven useful [1, 2].
- There is also a recent paper which discuss (out-of-domain) generalization in dynamical systems reconstruction [3] which might make sense to add to related work, as it also introduces measures to assess generalization with a focus on evaluating invariant measures of the underlying dynamics and addresses the problem that classical ERM framework is not enough to assess generalization in dynamical systems reconstruction
- While it is good to see that the Jacobian matching loss improves the reconstruction of invariant measures of the underlying dynamical system, it’s application in practice is fairly limited as Jacobians have to be estimated from data if the ground truth vector field is not known.
- I see that the authors want to study the case where training is only perform using “one-step-ahead” predictions (i.e. no unrolling of dynamics), however, I think considering the case of unrolling dynamics as a comparison to the Jacobian matching loss would be helpful as 1) unrolling dynamics is often enough to get a good reconstruction of the invariant measures of the observed attractor and 2) it is easy to apply in practice (no need for extra knowledge of the Jacobians). I.e. I think the more interesting question is whether knowledge of the Jacobians outperforms unrolling of dynamics during training significantly
- I think section 4, i.e. dynamic generative models, could be written a tad clearer, i.e. more explicit in terms of what exactly is now assumed to be stochastic in the specific VAE architecture and which part is deterministic. Maybe a Figure demonstrating the architecture setup in favor of moving e.g. proof of Theorem 1 into the Appx.
- To me the paper lacks a clear connection of the theoretical findings to the practical real-world setting, i.e. how these insights influence how we should train models to reconstruct real-world dynamics

I really like the idea and insights of this paper, but given the Weaknesses above, I do think it lacks actual practical relevance and the novelty is also very much limited to the (seemingly impractical) theoretical findings.

Miscellaneous comments:
- Figure 2 readability could be improved by removing the legend from within the plots and moving it to the outside (e.g. right, or between the plots), as the legend is the same for both plots
- Hyperlink of Table 3 jumps to section 3
- typo l. 192: “shdaowing”

[1] [Mikhaeil et al. (NeurIPS, 2022), On the difficulty of learning chaotic dynamics with RNNs](https://proceedings.neurips.cc/paper_files/paper/2022/hash/495e55f361708bedbab5d81f92048dcd-Abstract-Conference.html)

[2] [Hess et al. (ICML, 2023), Generalized teacher forcing for learning chaotic dynamics.](https://proceedings.mlr.press/v202/hess23a.html)

[3] [Göring et al. (ICML, 2024), Out-of-domain generalization in dynamical systems reconstruction.](https://arxiv.org/abs/2402.18377)

**Questions:**

- how exactly are the empirical measures $\mu$ and $\mu_{NN}$ and their $W_1$ distance computed (for Table 1 e.g.)?
- Could the authors include comparisons of the Jacobian matching loss to the more common training method of unrolling the dynamics for at least some time $t$?
- How can one draw practical consequences from the theoretical findings?

**Limitations:**

The authors do mention limitations of their approach (e.g. that the Jacobian matching loss is hard to implement in practice when ground truth is not known), but do not address how their findings might be used in real-world data settings.

---

> ### Author Rebuttal · Authors · 2024-08-06
>
> ### Related work on invariant measures
>
> Thank you for sharing these references -- we have added them to Related Work, which has spilled into the Appendix! Since the focus is on deriving dynamics-aware generalization bounds, we consider a minimal regression setup for one-step dynamics, which does not require specialized training methods such as the one introduced in [1]. In future work, it will be interesting to derive bounds similar to Theorem 2 for the sparse teacher forcing in [1] and other RNN/ESN-based training methods, as you point out. Many thanks for reference [3] that we have been remiss to overlook. Our setting is that of unique, ergodic, invariant physical measures in this paper, which is less complicated than multiple ergodic basins considered in [3]. It is reassuring that generalization with MSE loss does not imply statistical accuracy for the non-ergodic case considered in Sec 3.2 of [3]. Another interesting connection would be to specialize notions of generalization (Definition 3.2 specifically) introduced in [3] to the unique SRB/Gibbs measure case. While this notion or the W1 distance can be used to derive ERM problems, the sample complexity of these problems may be large. From the theoretical standpoint, we will extend our shadowing-based results to the non-ergodic case by considering weaker notions of finite-time shadowing in every connected component of the attractor. Different from [3], we demonstrate that even when we study ergodic systems, and sample from the entire state space, the traditional MSE does not yield the correct notion for generalization.
>
> ### Practicality of Jacobian loss and implications of theory
>
> This is indeed an important point, thank you and Reviewer PvwR for pointing this out! The computation of the Jacobian matrix estimated via automatic differentiation or via finite differences scales quadratically with the problem dimension. However, as illustrated in our numerical results, the sample complexity is indeed much smaller than for directly learning the physical measure via a generative model. Moreover, since the training strategy is an elementary regression on any simple architecture (like an MLP), it can be competitive when compared to more sophisticated training algorithms with recurrent architectures or transformers. In practice, whether to choose simple regression or to resort to transformers or recurrent architectures with innovative training stabilization strategies will depend on the dimension and the dynamical complexity (attractor dimension, number of positive Lyapunov exponents, correlation decay time etc). The computational complexity of complicated models such as transformers during inference time is quadratic in the dimension as well, making an elementary architecture appealing as a surrogate model.
>
> Furthermore, our result that adding Jacobian information implies statistical accuracy invites the development of practical supervised learning methods in high dimensions. Even though the theoretical results are derived for $C^1$ generalization, we are currently working on incorporating random Jacobian-vector products into the loss function. Such matrix vector products are easily tractable in several scientific applications where adjoint/tangent equation codes are available (see e.g geophysical turbulence models). One subject of our future work is to evaluate the efficacy of black box Jacobian-vector products in learning the physical measure.
>
> ### Unrolling dynamics
>
> We have run experiments on the Lorenz '63 system with an unrolled loss function: $\ell_{\rm u}(x) := (1/k) \sum_{t \leq k} \|v_t(x)\|^2, $ where $v_t(x) = F^t_{\rm nn}(x) - F^t(x)$ are vector fields on learned orbits. With $k = 10$ timesteps of unrolling time, we see that the [attractor is reproduced well](https://ibb.co/YBKhPjT), but [atypical orbits are still produced](https://ibb.co/dr9bPr6) for random initializations. The learned LEs are still incorrect: $[ 0.905, -0.03, -5.6]$ (the true LEs are :$[0.85, 0, -14.5].$) When the unrolling time $k$ is increased, we still obtain the same results, with even the positive LE being overestimated.
>
> Overall, unrolling seems to learn more accurate representations than the MSE model but worse than the JAC model. We also observe that the unrolling time $,k,$ needs to be fine-tuned as a hyperparameter to achieve good generalization; a small perturbation in $k$ can lead to training instabilities. We have these results to Appendix C of our revision.
>
> To understand these results, for short times, $t,$ (when compared to the Lyapunov time, $1/\lambda_\mathrm{max}),$ $v_t$ are in tangent spaces along the NN-orbit, $\{F^t_\mathrm{nn}(x).\}$ Now, we note that $v_t$ is the pushforward through the Jacobian/linearized dynamics of $v_{t-1},$ which yields the recursive relationship, $v_t(x) = dF(F^t(x)) v_{t-1}(x) + v_0(F_{\rm nn}^t(x)).$  Thus, the unrolled loss does contain Jacobian information implicitly although it does not enforce the learned trajectory to be close to the true trajectory in $C^1$-distance. This serves to explain, via Theorem 2, why it works better than the MSE (even in some practical climate emulators, e.g., FourcastNet). Thank you very much for raising this important point, which supports the central argument of this paper: adding Jacobian information leads to statistical accuracy.
>
> ### W1 distance
>
> We use Python Optimal Transport package, which uses the Sinkhorn algorithm. For our 1D distributions, the optimal transport map is analytically the target inverse CDF $\circ$ CDF of source, which is estimated via projected gradient descent in the package.
>
> ### Edits
>
> Many thanks for a careful reading and suggesting many improvements (implemented in the revision, including adding limitations to sec 7) to the paper! We have rewritten ``Dynamic generative models'': here, the dynamics $\Phi^t_\phi$ is stochastic, while the encoder and decoder are deterministic. This is needed for greater expressivity of the conditional measures.

---

> > ### Comment · Reviewer_vAp3 · 2024-08-12
> > **Re: Rebuttal**
> >
> > > Related work on invariant measures
> >
> > Happy to hear that the authors add the references to the related work and thanks for the additional clarification and connection to their own work!
> >
> > > Practicality of Jacobian loss and implications of theory
> >
> > Indeed, the complexity of computing the *model Jacobian* using AD/Finite differences is one part, however, did the authors also consider the added complexity by including this Jacobian in the loss function, which means one has to a backward-over-forward AD problem (-> second order differentiation)? Does runtime/complexity still only scale with problem dimensionality squared?
> >
> > My main concern with the practicality is the estimation of the *Jacobians from time series data*, where **no** ground truth knowledge is available (I appreciate the geophysics examples and I see that in these cases the Jacobian matching loss might improve things). I really miss any experiments on trying to make this loss more applicable in real-world scenarios (e.g. estimating Jacobians from time delay embeddings of real-world data -> do crude approximations already help learning dynamical invariants?).
> >
> > > Unrolling dynamics
> >
> > Thanks for conducting the additional experiments! That zero/negative LEs are still underesimated when unrolling dynamics is indeed a problem I encountered in practice, too. Hence it is great to see that knowledge of the Jacobians can eliminate this problem. However, I checked the paper again and found that the authors report a ground-truth max. LE for the Lorenz63 system with standard settings of ~0.86 (cf. Table 2). However, to my knowledge the Lyapunov spectrum is given by ~[0.905, 0, -14.57]. I checked this with minimal code using Julia (see below). Do the authors mind to share how they estimated the ground-truth Lyapunov spectrum or whether they used specific literature values?
> > ```
> > using DynamicalSystems, ChaosTools
> >
> > # Lorenz63
> > ds = Systems.lorenz(ρ=28.0, σ=10.0, β=8 / 3)
> >
> > # Lyapunov spectrum
> > lyap = lyapunovspectrum(ds, 100_000, Ttr=1000) # ~ [0.905, 6e-6, -14.57]
> > ```
> > > W1 distance
> >
> > Thanks for the clarification!
> >
> > > Edits
> >
> > I see, thanks!

---

> > > ### Author Response · Authors · 2024-08-12
> > > **Jacobian and LEs**
> > >
> > > Thank you very much for reading through the rebuttal! We really appreciate your interest and excellent questions!
> > >
> > > > Does runtime/complexity still only scale with problem dimensionality squared?
> > >
> > > This is a very keen observation -- thank you. We meant that computing the Jacobian (using Finite Difference or AD) has quadratic complexity, but you are absolutely right that training time increases also because of differentiating the model at the training points. During training the Jacobian (with respect to the input state) is differentiated with respect to the parameters of the neural network. The complexity with respect to the dimension of the problem still remains quadratic but the training complexity increases just like a physics-informed neural training; the factor of increase therefore also depends on the complexity of the network. Inference time is the same as a vanilla neural ODE however since only the training is modified. Practically we found that KS system (128-dimensional system) was still training in about 10 hours on a single RTX30xx GPU.
> > >
> > > > estimation of the Jacobians from time series data, where no ground truth knowledge is available
> > >
> > > This is exactly the scenario we had in mind for the rebuttal. We can estimate the Jacobian from timeseries data using finite difference as long as the data sample the system frequently (there are classical results known for the impossibility of system identification for longer observation frequency and very noisy observations of chaotic systems). As you correctly point out, there is an error made by finite difference (which is on order of epsilon^2, where epsilon is the step size norm). However, even though our theoretical results only use exact Jacobians, the results with unrolling dynamics, e.g., where only indirect Jacobian information is available, suggest that even inexact Jacobians can help us learn the dynamical invariants better.
> > > For instance, even Lyapunov vectors and exponents can be estimated with high accuracy using just finite difference estimates of the Jacobian (this is easy to try by replacing the AD with inexact Jacobians in the code snippet below).
> > >
> > > But as you suggest, we should test this on real world high-dimensional examples (even ones where the attractor is only known through delay embedding), which we defer to a future work. Our present paper emphasizes on understanding why Jacobian information leads to statistical accuracy. Our main contribution is using shadowing theory to provide an explanation and suggest that there is hope of learning chaotic systems with elementary supervised learning methods, and that these can still learn the invariant measure.
> > >
> > > About the Lyapunov exponents, small differences (statistical error/noise) may arise depending on the integration time (len(traj_gpu) in the code below), time integration method (RK4 in the code below), and subsequently how we estimate Jacobians. Here is the code snippet we wrote for computing 3 LEs:
> > >
> > > ```
> > > def lyap_exps(dyn_sys_info, traj, iters):
> > >     model, dim, time_step = dyn_sys_info
> > >     LE = torch.zeros(dim).to(device)
> > >     traj_gpu = traj.to(device)
> > >     f = lambda x: rk4(x, model, time_step)
> > >     Jac = torch.vmap(torch.func.jacrev(f))(traj_gpu)
> > >     Q = torch.rand(dim,dim).to(device)
> > >     eye_cuda = torch.eye(dim).to(device)
> > >     for i in range(iters):
> > >         if i > 0 and i % 1000 == 0:
> > >             print("Iteration: ", i, ", LE[0]: ", LE[0].detach().cpu().numpy()/i/time_step)
> > >         Q = torch.matmul(Jac[i], Q)
> > >         Q, R = torch.linalg.qr(Q)
> > >         LE += torch.log(abs(torch.diag(R)))
> > >     return LE/iters/time_step
> > > ```
> > >
> > > Thank you very much for your questions and please feel free to let us know if anything is unclear!
> > >
> > > Thank you again for your time and help!

---

> > > > ### Comment · Reviewer_vAp3 · 2024-08-13
> > > > **Re: Jacobian and LEs**
> > > >
> > > > Thanks for the clarification - makes sense that the runtime complexity increases similar to PINN training!
> > > >
> > > > Regarding the LEs; First of all, thanks for providing the code. I think this needs more careful evaluation. If you call something "true" LEs, you should indeed compute the right quantity - for the true spectrum, the statistical error or noise should be negligible. I.e. you need to let the trajectories converge into their limit set and draw a long enough trajectory to pin down the correct spectrum (you may plot time steps/integration time vs error in the spectrum to verify you draw long enough orbits to converge). But if the reported "true" LE is indeed ~0.905 instead of ~0.85, then this changes the interpretation of Table 2 (is it just that the models need to be evolved for longer or are they then indeed underestimating the max. LE even with Jac. loss?).
> > > >
> > > > To be clear, I really appreciate the effort the authors put into the paper and rebuttal and I think the paper is of great value, but to me it seems it could really benefit from another revision and submission at a later point. I will raise my score to 5 with the assumption that the authors will include all additional experiments in the final version (reevaluate LE spectrum calculation, making sure the true spectrum is reported, estimating Jacobians from data alone, addressing computational complexity).

---

> > > > > ### Author Response · Authors · 2024-08-13
> > > > > **Re: Jacobian and LEs**
> > > > >
> > > > > Thank you for reading through our comment and for your response! We are so grateful to you for responding promptly and thoughtfully!
> > > > > You are absolutely right. Since LEs are statistical quantities, and for a singular hyperbolic system like the Lorenz system, the Central limit theorem holds, we do see a 1/sqrt{N} convergence of ergodic averages and LEs computed along trajectories of N length. The "true" label does refer to long averaging times conducted with the ground truth system -- i.e., the true vector field. It is indeed a fair comparison to maintain an equally long time averaging but compute the LEs with the Neural ODE as a vector field instead. These are what are referred to as "JAC" or "MSE" values -- the LEs computed with these learned models. A 5% difference in LEs can happen even in between runs with different initial conditions and this should, for instance, not be considered as an indication of lack of ergodicity. Overall, we want to clarify that the LEs are expectations of random variables and therefore cannot be estimated like deterministic quantities. It would not be meaningful to report them (in a chaotic system) with many digits of precision, e.g.
> > > > > Please let us know if any further clarification is needed!

---

### Official Review · Reviewer_L3o7 · 2024-07-12

**Soundness:** 3
**Presentation:** 3
**Contribution:** 3
**Rating:** 6
**Confidence:** 2

**Summary:**

This paper extends generalization results to models trained on dynamical data, especially Neural ODEs. The paper shows and then attempts to explain why Neural ODE trained without a Jacobian matching term fail to capture physical behaviour even when they have low generalization error. Under a generalization assumption, the paper shows that the Jacobian-matching loss can lead to statistically accurate models. The paper shows a number of experiments that show the behaviour stipulated by the theoretical results.

**Strengths:**

The paper is technical but well written and ideas are clearly explained.

The mathematical formulation is quite clean and targets an important problem of generalization in ODE models for dynamical data.

The explanation of generalization for Neural ODEs is I think novel.

**Weaknesses:**

I think the main weakness is assumption 1. There should be some explanation of the difference between C1 and strong C1 generalization. Furthermore, there should at least be some justification of why this should hold and when we could not expect it to hold.

Another possible problem I see is this: The paper explains why minimizing loss 3 (with the jacobian) implies statistical accuracy. I don’t think that it properly explains why minimizing loss 2 does not imply statistical accuracy. I would understand why a standard MLP might not match derivatives. But I have a harder time understanding why a neural ODE with low generalization error isn’t able to do that. Some more intuition about this would be useful.

**Questions:**

Related to the above. In lines 264-265 the paper states that C0 generalization is insufficient for learning shadowing orbits. Is it an obvious fact? Further explanation would be useful. Also useful would be a comparison of a standard MLP minimizing loss 2 and a neural ODE minimizing the same loss.

**Limitations:**

The paper does not have a section on limitations. I think a discussion of limitations would add to the value of the paper.

---

> ### Author Rebuttal · Authors · 2024-08-06
>
> ### When Assumption 1 is expected to hold
>
> A necessary condition for Assumption 1 ($\mathcal{C}^1$ strong generalization) is that the optimization problem with the Jacobian loss is solved "well" -- resulting in low generalization errors. But, as you have carefully observed, this is insufficient to claim this notion of strong generalization. The Jacobian loss must be small along orbits of the learned NN (NN with small Jacobian generalization error). Note that these orbits are $(\epsilon_0, \epsilon_1)$ orbits of the true system. When these orbits are on or near the support of $\mu$ (the physical measure of the true system), then, the Jacobian loss is small at those points, and $\mathcal{C}^1$ strong generalization holds. Thus, a sufficient condition is for the true dynamics to be such that small $\mathcal{C}^1$ perturbations of it lead to small perturbations of the physical measure. Such a condition is called smooth linear response in the parlance of dynamical systems and has been extensively studied in the theoretical literature (e.g., see the review by Baladi here: https://arxiv.org/abs/1408.2937). For uniformly hyperbolic systems, which are considered in this paper, and many chaotic systems observed across physics, such a smooth linear response holds (https://iopscience.iop.org/article/10.1088/0951-7715/22/4/009/meta). But, of course, this is only a sufficient condition, and in practice, Assumption 1 may be satisfied more easily. For instance, when we train by sampling points randomly in a box around the attractor, we automatically sample from points near the support of $\mu$ (the attractor). Thus, in input space regions where orbits of $F_{\rm nn}$ live, the Jacobian loss might evaluate to a small value. That is, whether or not linear response holds, we achieve $\mathcal{C}^1$ strong generalization by choosing training points randomly distributed (according to any density) around the attractor and minimizing the Jacobian loss. We thank you for this excellent question, which has led to an important clarification in the paper. Due to space constraints in the paper, we have added more details in the appendix and only briefly make a clarification after Assumption 1 in the main text.
>
> ### Why minimizing MSE loss does not yield statistical accuracy
>
> This is an astute observation, thank you. Indeed the fundamental contribution of the paper is to suggest shadowing as a mechanism for learning the physical measure. Shadowing does not hold when we only learn well with respect to the $\mathcal{C}^0$ loss! This is because in order to shadow a chaotic orbit, informally, we need to predict the next step as well as local directions of infinitesimal linear perturbation (linear/Jacobian structure) induced by the next step dynamics. This intuition is implicit in the proof of the shadowing lemma (see e.g., Chapter 18 of Katok and Hasselblatt), and therefore extends to the proof of our high-probability version of the shadowing lemma that leads to our main result. Just learning the short-term dynamics without learning the local directions of growth/decay of infinitesimal perturbations cannot lead to learning shadowing orbits. Since we prove that shadowing is the underlying mechanism for learning the invariant measure, the MSE loss is insufficient. We hope this clarifies our result -- thank you for the great question! Due to space constraints, we have not added this explanation, but we will in the next revision right after Theorem 2.
>
> ### Neural ODEs are neural parameterizations of the vector field
>
> Neural ODEs (trained with MSE) learn the vector fields, but we need to learn the derivatives of the vector fields (with respect to the state vector) for statistical accuracy.
> Our results indeed already compare Neural ODEs where the vector field is parameterized by an MLP, MLP with Fourier Layers and ResNet blocks. Our observation is the same: without the Jacobian information in the loss function, even parameterizing the vector field, like a Neural ODE does, is not sufficient to learn physical measure.
>
> Intuitively, one can argue that learning more and more derivatives of a function (like a vector field of a flow or a discrete-time dynamical system, $F$) would lead to more accurate learning the function and hence also give rise to statistical accuracy. But shadowing (which only requires first derivatives) says that higher-order derivative matching is not necessary.
>
> ### Limitations
>
> We add the discussion on limitations in Section 7 of the revision. Mainly, we illustrate our theoretical results using the full Jacobian, but this does not yield a practical scheme for training. Secondly, our results are derived for mathematically ideal chaotic systems trained with a vanilla regression setup. Training interventions that enforce learning invariant measures are not considered in our analysis. In this regard, please see our responses to Reviewer vAp3 and sRDx.

---

> > ### Comment · Reviewer_L3o7 · 2024-08-13
> >
> > Thank you for the clarifications.
> > From the other reviews, I see there are some concerns regarding practicality of obtaining the Jacobian. However, I think the explanation of generalization also has value. I maintain my accept recommendation.

---

> > > ### Author Response · Authors · 2024-08-13
> > > **Thank you!**
> > >
> > > Thank you very much -- we especially appreciate your questions!
> > > The purpose of this paper, as you indeed correctly point out, is dynamics-aware generalization bounds. We show that learning the Jacobian leads to statistical accuracy and *explain why* by deriving a generalization bound via shadowing theory.
> > > Computing the Jacobian is actually not impractical compared to more sophisticated training approaches using recurrent architectures or generative modeling of the SRB measure (physical measure in the paper).
> > > We appreciate your time and effort! Please let us know if we can provide any further clarification!

---

### Official Review · Reviewer_sRDx · 2024-07-16

**Soundness:** 2
**Presentation:** 3
**Contribution:** 3
**Rating:** 7
**Confidence:** 4

**Summary:**

The authors focus on analyzing why MSE loss fails to capture the physical behavior of dynamical systems. Narrowing their analysis to invariant ergodic systems, they provide theoretical insights on when generalization implies statistical accuracy. They propose that for models to be statistically accurate, they must reproduce dynamical invariants, which is achieved by ensuring that the learned model closely follows the true system's orbits. Specifically, the authors provide theoretical justification for why Jacobian matching loss can better capture statistical properties than MSE loss. Empirically, they verify their analysis on the Lorenz 63 system using different architectures, including MLP, Neural ODE, and FNO.

**Strengths:**

The authors provide a thorough theoretical analysis and propose some useful notions to characterize models' ability to reproduce the dynamics statistical measure. Overall, the paper is well-written and easy to follow.

**Weaknesses:**

1. Jacobian loss (Eqn.3) considered in the paper is a special case of the Sobolev norm in [1]. The authors should consider extending their analysis to the Sobolev norm.

2. Recent related works have also discussed the problem of MSE and shown multiple ways to improve upon MSE loss, e.g., the Wasserstein loss used in [2,3] and the theoretical analysis provided in [4]. Although some of these works might focus more on the empirical side, my understanding is that this problem has raised more attention than the authors' claim, and some comparison to the existing work is necessary.

[1] Learning Dissipative Dynamics in Chaotic Systems (https://arxiv.org/pdf/2106.06898)

[2] DySLIM: Dynamics Stable Learning by Invariant Measure for Chaotic Systems (https://arxiv.org/abs/2402.04467)

[3] Training neural operators to preserve invariant measures of chaotic attractors (https://arxiv.org/abs/2306.01187)

[4] On the difficulty of learning chaotic dynamics with RNNs (https://arxiv.org/pdf/2110.07238)

**Questions:**

Q1: Figure 1 is unclear, especially when the y-axis is not aligned in the 4&5 columns, and the message shown in Figure 1 seems to contradict the authors' theoretical claims, as the probability distribution of the modeled dynamics is still mismatched when the dynamics of the attract seems to be well learned.

Q2: It's confusing when the authors interchangeably used notations $h$ and $F$ for the dynamics map. Is there a particular reason for switching between these notations, and could you elaborate on why?

Q3: Line 183: It's not clear why the authors state that implementing Wasserstein distance is difficult when the map $h$ is chaotic.

Q4: As Lyapunov exponents are calculated using local dynamics, could you show a comparison using some long-term evaluation metrics?

---

> ### Author Rebuttal · Authors · 2024-08-06
>
> ### Implications for training with Sobolev norm
>
> It is indeed interesting to consider errors in the learned dynamics as distributions in a Sobolev space, $W^{k,p}$, as you point out. You are absolutely correct that our MSE loss is a special case for $k = 0, p =2$ and the Jacobian loss for $k= 1, p =2.$
> Here, we only consider classical functions (as opposed to distributions) for the true map $F$; the neural network, $F_{\rm nn},$ also does not have singularities since we use a smoothed out ReLU as activation.
>
> Hence, the weak derivatives above are indeed just classical derivatives. Our results (Theorem 2) means the following for the Sobolev loss: we do not gain more statistical accuracy by minimizing errors with larger $k,$ that is, by considering higher-order derivatives. When $p=2,$ we do not gain by including in the error or loss term higher order Fourier coefficients. In this paper, we prove that a sufficient condition for statistical accuracy is the prevalence of shadowing and its typicality (that is, distribution of shadowing orbits according to $\mu$ with high probability). Our high-probability version of shadowing only requires $C^1$ strong generalization: that is, matching derivatives only up to first order. Therefore, when a shadowing based mechanism for generating the physical measure holds in a supervised learning problem, our results imply that matching higher-order derivatives are not necessary.
> We thank you for making this important suggestion! Due to space constraints, we have not added this to the revision but plan to add a more detailed analysis to the next revision.
>
> ### Related work
>
> These are valuable additions to our related work, we are greatly indebted to you and Reviewer vAp3 for showing us these references! We have added all of the references in the Related Work and continued the section into an appendix due to lack of space. Regarding the first cited reference, it is very interesting to derive generalization bounds for the dissipativity-enforcing loss proposed by the authors. Without enforcing dissipativity, in our work, we find that incorporating the Jacobian learns an attractor of zero volume (the attractor dimension is related to the LEs and these are correctly obtained) in dissipative systems.  We do not consider training interventions as done in [2] and [4] to stabilize the training of chaotic dynamics over longer time horizons. We leave performing theoretical analysis of the statistical accuracy of the empirical approach to train RNNs in [4] for future research. Instead, our focus here is on obtaining provable guarantees for learning of the physical measure; hence, we only consider here the minimal training setting of regression over short time horizons, wherein such stabilization interventions are not needed. The reference [3] is indeed very close in motivation to our work in that invariant measures of chaotic systems are learned by neural network models. However, the approach taken in [3] is markedly different since optimal transport is performed on key statistics or such summary statistics are approximated through contrastive learning. We argue that supervised learning with Jacobian information is less complicated and more tractable in high dimensions even when compared to an efficient OT algorithm (Sinkhorn -- on discrete measures -- is used in the reference). Furthermore, we obtain theoretical guarantees for the error in the learned invariant measure (e.g. in Wasserstein distance), while it would be interesting to derive such guarantees for the contrastive learning approach taken in [3].
>
> ### Clarification of Figure 1
>
>
> The y-axis of the 4th and 5th columns of Figure 1 were the empirical PDFs of a random orbit generated by the MSE and Jacobian models respectively. The different scale of the plots show that the empirical distribution of the MSE orbit is incorrect. This exemplifies the thesis of the paper: adding Jacobian information in the regression problem leads to learning the physical measure. To avoid confusion, we have now [combined the 4th and 5th plots](https://app.gemoo.com/share/image-annotation/679582047652405248?codeId=vzaQe6gZzGngO&origin=imageurlgenerator&card=679582046868070400) into one figure, revised the caption and added a Gist (please see the link). Thank you!
>
>
> ### Clarifying notation of map
>
> We apologize for the confusion: the h is used as an argument for writing the generalization error as a function of the model $h$. The learned chaotic map is denoted throughout as $F_{\rm nn}$ and the true map by $F$.
>
> ### Solving OT problems with ergodic measures of chaotic maps
>
> Thank you for the careful observation! Suppose we are trying to minimize the Wasserstein distance, whose dual form we can lower bound, for some $g \in {\rm Lip}^1(M),$
> as
> $$
> W^1(\mu, \mu_{\rm nn}) \geq \lim_{t\to\infty} \dfrac{1}{t} \sum_{n\leq t} |g(F^n(x)) -  g(F^n_{\rm nn}(x))|,
> $$
> for Leb a.e. x. In practice, e.g., when solving OT problems with the Sinkhorn algorithm, we replace the continuous measure with a discrete measure, which in the above case reduces to replacing the ergodic average ($t\to \infty$) with an average over an orbit of finite length. Then, the estimate of the above error is noisy when $F$ and $F_{\rm nn}$ are chaotic, and indeed the variance grows exponentially with $t$ and then saturates (due to both attractors being bounded).
>
> ### LEs are statistical measures
> LEs are indeed long-term evaluation metrics (now added in Section 2). For a vector field $E_i$ in the $i$th Oseledets subspace, we can rewrite the definition of the $i$th LE as an ergodic average: $$\lambda_i := \lim_{t \to \infty} \dfrac{1}{t} \sum_{n =1}^t \log \|dF (F^n(x)) E_i(F^n(x))\|.$$
> We also show empirical distributions of various quantities (like the components of the state vector) estimated over the long orbits directly in Figure 1 and Table 1. Thank you for this suggestion!

---

> ### Comment · Reviewer_sRDx · 2024-08-14
> **Response to your rebuttal**
>
> Thank you for your response! Your response regarding the Sobolev norm was valuable, and your further clarifications cleared up my concerns. I appreciate your discussion regarding the related works, which helped me better assess the status of your work. I will increase my score to accept.

---

> > ### Author Response · Authors · 2024-08-14
> > **Thank you!**
> >
> > We really appreciate your time and for re-assessing our work based on our rebuttal!
> > Thank you!

---

### Author Response · Authors · 2024-08-06
**Thank you for the feedback**

Dear Reviewers,

We are extremely grateful to you for spending the time to understand our paper and raise many interesting questions. Due to lack of space, we have not quoted from your review in our responses, but we have answered all your comments to the best of our knowledge under easily identifiable headings. We outline the revisions/edits that we have already made to our paper, and as soon as the option becomes available, we will upload the revised version. For new experiments that we have conducted, we have shared **anonymous links to the figures** since we cannot yet share the revised paper. We request you to visit these links to see the plots for the new experiments on unrolled loss functions. We have indicated in our response wherever there are similarities in the questions/comments by more than one Reviewer. We request you to therefore read the *relevant sections of the rebuttal to other reviewers* as well.
We thank you once again for your time and valuable feedback that has improved our paper substantially!

---

### Author Response · Authors · 2024-08-11
**Request for response**

Dear Reviewers,

We request you to please find time to read our rebuttal and help us identify any concerns that we have not yet addressed.
We really appreciate your time and several insightful comments, which have considerably enhanced our paper!

---

### Author Response · Authors · 2024-08-13
**Request to read rebuttals**

Dear Reviewers,

Since the discussion period is set to close soon, we request you to please read our rebuttals and ask for clarification. If we have answered all your questions satisfactorily, we request you to please re-evaluate our paper.

---

### Author Response · Authors · 2024-08-14
**Thank you and request for response!**

Dear Reviewers,

Thank you for reading our rebuttals, seeking further clarification and re-evaluating our paper!
We are extremely grateful for all the time you have spent on improving our work.
If you have not done so already, could you please read our rebuttal, and re-assess our work in light of the clarifications in the rebuttal?
Sorry for the multiple reminders -- we are almost at the end of the discussion period.

Thank you again!

---

### Author Response · Authors · 2024-08-14
**Thank you for the comments and discussion on learning chaotic dynamics**

Dear Reviewers and ACs,

We have truly grateful for all the time you have spent in giving us a thorough review. The subsequent discussions (including the references) have been very fruitful and led to substantial improvements in the revision. We are excited about both directions our work contributes to and is influenced by: practical methods for learning physical measures (ensemble/long-term behavior) of chaotic dynamics using supervised learning and introducing dynamical systems/ergodic theory to analyze and understand generalization.  Overall, the intersection of nonlinear dynamics with machine learning theory and practice is growing and we are very happy for the opportunity to be able to improve our work at this intersection through these discussions with you.

---

### Decision · Program_Chairs · 2024-09-25

**Decision:**

Accept (poster)

**Comment:**

All reviews agreed on the premise of the paper: existing approaches to learning dynamical systems using MSE losses fail to generalize, and the paper introduces a Jacobian-matching loss that can learn underlying properties such as orbits and invariants. The major weakness identified by two reviewers is that the Jacobian is often not possible to get from data and some other alternative signal would be needed in most practical situations. The discussion resulted in extra experiments that other practical alternatives still do not perform as well as the Jacobian-matching method, and there was agreement that the proposed method is still a valuable contribution. All reviewers agreed on the theoretical soundness of the paper.

After discussion, all the reviewers all agreed that this paper should be accepted. One reviewer was borderline, but after the discussion, the authors promised to add an additional experiment in the revision:
> “ I think the paper is of great value ...I will raise my score to 5 with the assumption that the authors will include all additional experiments in the final version (reevaluate LE spectrum calculation, making sure the true spectrum is reported, estimating Jacobians from data alone, addressing computational complexity).”

The other three reviewers did not think that these results were necessary for their accept recommendations. The AC thinks the additional results obtained in the discussion period would be easy to include in the revision and not dramatically change the paper. As such, the paper can be accepted.